# Hyperphosphorylated PTEN exerts oncogenic properties

**Janine H. van Ree**[1], **Karthik B. Jeganathan**[1], **Raul O. Fierro Velasco** [1],
**Cheng Zhang**[2], **Ismail Can** [3], **Masakazu Hamada**[1], **Hu Li** [2], **Darren J. Baker** [1,3] &
**Jan M. van Deursen** [1,3] ✉

PTEN is a multifaceted tumor suppressor that is highly sensitive to alterations in expression or function. The PTEN C-tail domain, which is rich in phosphorylation sites, has been implicated in PTEN stability, localization, catalytic activity, and protein interactions, but its role in tumorigenesis remains unclear. To address this, we utilized several mouse strains with nonlethal C-tail mutations. Mice homozygous for a deletion that includes S370, S380, T382 and T383 contain low PTEN levels and hyperactive AKT but are not tumor prone. Analysis of mice containing nonphosphorylatable or phosphomimetic versions of S380, a residue hyperphosphorylated in human gastric cancers, reveal that PTEN stability and ability to inhibit PI3K-AKT depends on dynamic phosphorylation-dephosphorylation of this residue. While phosphomimetic S380 drives neoplastic growth in prostate by promoting nuclear accumulation of β-catenin, nonphosphorylatable S380 is not tumorigenic. These data suggest that C-tail hyperphosphorylation creates oncogenic PTEN and is a potential target for anti-cancer therapy.

The *PTEN* tumor suppressor gene is mutated in the germline of patients with PTEN hamartoma tumor syndrome (PHTS) and in a high proportion of sporadic cancers[1–4]. In addition to biallelic *PTEN* mutations, human cancers often show loss of only a single *PTEN* allele characteristic of haploinsufficient tumor suppressor genes[5,6]. Partial loss of PTEN can be a driver of tumorigenesis, which was suggested by the graded reduction of PTEN levels in mice, with PTEN dosage inversely correlating with tumor susceptibility[7–9]. Even subtle reductions in PTEN level increase tumor susceptibility in these experiments, highlighting that faithful regulation of PTEN expression and activity are critical for tumor suppression[10,11]. Mechanisms regulating PTEN expression and activity are complex, where deregulation can occur in various ways, including epigenetic silencing, transcriptional repression, microRNA regulation, interruption of competitive endogenous RNA networks, aberrant post-translational modification, and subcellular mislocalization[12–15]. In addition, PTEN homodimerizes and has many binding partners that modulate its function[16,17], further

adding to the notion that PTEN is a tumor suppressor with numerous vulnerabilities.

Insights into the regulation and functions of PTEN mostly involve studies in cultured cells. Extending these studies at the organismal level to demonstrate relevance to tumor suppression is particularly challenging because PTEN is essential for mouse development, with embryogenesis halting around day 9.5 in *Pten*[−/−] mice[18–20]. Various mutations found in PHTS patients have been successfully modelled in mice, with heterozygous mutants typically showing cancer predisposition and embryonic lethality in homozygotes[17,21]. This includes missense mutations in the phosphatase domain, reinforcing that negative regulation of the oncogenic PI3K-AKT-mTORC signaling network through dephosphorylation of the lipid signaling intermediate PIP3 is a major tumor suppressive function of PTEN[4,22]. The close correlation between AKT hyperactivity and increased tumor susceptibility in *Pten* mouse models further supports this notion[7–9,17,23]. However, several mouse strains modeling *Pten* mutations found in

[1]Department of Pediatric and Adolescent Medicine, Mayo Clinic, Rochester, MN, USA. [2]Department of Molecular Pharmacology and Experimental Therapeutics, Mayo Clinic, Rochester, MN, USA. [3]Department of Biochemistry and Molecular Biology, Mayo Clinic, Rochester, MN, USA.
✉e-mail: janvandeursen2@gmail.com

human cancers develop tumor phenotypes in the absence of marked AKT alterations. This includes mice heterozygous for a C2 domain missense mutation at phenylalanine 341 or a C-terminal truncating mutation that causes chromosomal instability[24,25], as well as mice homozygous for a deletion of residues 401-403 (TKV) spanning the PTEN PDZ-binding domain (PDZ-BD). In-depth analysis of these $Pten^{\Delta TKV/\Delta TKV}$ mice uncovered that a centrosome-associated PTEN pool recruits Dlg1-Eg5 to duplicated centrosomes via the PDZ-BD to establish symmetrical bipolar spindles that properly segregate chromosomes[26]. PTEN accumulation at centrosomes requires phosphorylation of S380, a residue that is not a mutational target in human cancers but whose hyperphosphorylation is a hallmark of human gastric cancers[27–29]. S380 is located within the PTEN C-tail domain spanning amino acids 352-400 and part of a cluster of CK2 phosphorylated serine and threonine residues that also includes S370, T382, T383 and S385[14,30–35]. Ectopic expression of PTEN variants with mutations in these residues in cultured cells indicated that C-tail phosphorylation enhances PTEN stability while reducing membrane localization and phosphatase activity[14,31,33,36]. This seems paradoxical, but when phosphorylated, the C-tail interacts with the C2 and phosphatase domains to form a "closed" conformation. This conformer is less susceptible to ubiquitin-mediated degradation and caspase 3 cleavage, yet is less capable of interacting with proteins that target PTEN to the plasma membrane, resulting in it being less catalytically active[12,14,37,38].

In this work, we examined the physiological relevance of the PTEN C-tail domain and its phosphorylation status by introducing subtle, non-lethal mutations. We show that C-tail mutant mice with an S380A substitution or a deletion that includes S370, S380, T382 and T383, display low levels of PTEN and increased AKT signaling with no tumor predisposition. On the other hand, mice carrying an S380D phosphomimetic mutation exhibit β-catenin hyperactivity in addition to PTEN instability and AKT hyperactivity. We show this drives prostate neoplasia and have found support that this mechanism may be clinically relevant in gastric and prostate cancer.

## Results

### PTEN mutant S380A is unstable and causes chromosome missegregation

We used CRISPR-Cas9 gene editing in FVB fertilized eggs to introduce an S380A missense mutation in the endogenous Pten locus. In doing so, we also obtained mice containing an in-frame deletion of amino acids 369-383, which includes four C-tail phosphorylation sites, S370, S380, T382 and T383 (Fig. 1a and Supplementary Fig. 1a). We refer to the allele encoding this mutant as $Pten^{\Delta 4}$. $Pten^{S380A/A}$ and $Pten^{\Delta 4/\Delta 4}$ mice were born at Mendelian frequency from heterozygous crosses and were overtly indistinguishable from control littermates.

Western blot analysis of tissues and MEFs of $Pten^{S380A/A}$ (A/A) and $Pten^{\Delta 4/\Delta 4}$ (Δ4/Δ4) mice revealed that both PTEN$^{S380A}$ and PTEN$^{\Delta 4}$ are expressed at subnormal levels (Fig. 1b–e, Supplementary Fig.1b–d). In all tissues analyzed, PTEN$^{S380A}$ was reduced to the same or a slightly lower level than PTEN in corresponding tissues of $Pten^{+/-}$ (+/−) mice, except for lung. PTEN$^{\Delta 4}$ levels were always most profoundly reduced regardless of tissue (Fig.1c–e, Supplementary Fig. 1d). Cycloheximide chase experiments in MEFs indicated that low PTEN$^{S380A}$ and PTEN$^{\Delta 4}$ levels were caused by protein instability (Fig. 1f, Supplementary Fig. 1e). Complementary cell fractionation experiments on MEFs and prostates revealed that, in contrast to $Pten^{+/-}$ cells, nuclear PTEN in $Pten^{S380A/A}$ and $Pten^{\Delta 4/\Delta 4}$ cells was not reduced (Fig. 1g), whereas cytoplasmic and membrane-associated PTEN levels were markedly decreased regardless of genotype.

Detailed analysis of mitotic $Pten^{S380A/A}$ and $Pten^{\Delta 4/\Delta 4}$ MEFs revealed defects reminiscent of those observed earlier in $Pten^{\Delta TKV/\Delta TKV}$ MEFs[26] supporting the idea that S380 phosphorylation is required for DLG1-mediated centrosomal accumulation of PTEN (Supplementary Fig. 2

and Supplementary Note). However, because the same mitotic phenotypes have also been observed in $Pten^{+/-}$ MEFs[26], they could alternatively be due to the PTEN instability in $Pten^{S380A/A}$ and $Pten^{\Delta 4/\Delta 4}$ MEFs.

### PTEN C-tail mutant mice are not tumor prone despite AKT hyperactivity

Because PTEN deficiency is tightly associated with tumorigenesis, we expected C-tail mutant mice to be tumor prone. However, $Pten^{S380A/A}$ and $Pten^{\Delta 4/\Delta 4}$ mice showed no evidence of tumor predisposition at 9 months (Fig. 2a,b), an age where macroscopic tumors are known to be readily detectable in $Pten^{+/-}$ mice[18,19,21]. We reasoned that a possible explanation for this discrepancy might be that $Pten^{+/-}$ mice used in tumorigenesis studies were on a C57BL/6 or a C57BL/6 ×129 genetic background while our C-tail mutants were on a pure FVB background. To examine this further, we used CRISPR-Cas9 gene editing in FVB fertilized eggs to introduce an out-of-frame deletion in Pten exon 2, thereby creating a knockout allele (Supplementary Fig. 3a). When screened for macroscopic tumors at 9 months, the resulting FVB $Pten^{+/-}$ strain showed high tumor predisposition with lymphomas and pheochromocytomas as the most prominent tumors (Fig. 2a,b, Supplementary Fig. 3b). However, we did not detect a susceptibility to breast tumors on an FVB background. Most of the sacrificed $Pten^{+/-}$ mice exhibited lymphadenopathy (Fig. 2c), a prominent non-neoplastic phenotype previously reported for $Pten^{+/-}$ mice on a C57BL/6 or a C57BL/6 ×129 genetic background and typically requiring euthanasia before 12-14 months of age. Lymphadenopathy was not observed in C-tail mutant mice which allowed for complementary assessment at 16 months for tumors with longer latencies, but again there were no differences in tumorigenesis between $Pten^{S380A/A}$, $Pten^{\Delta 4/\Delta 4}$ and $Pten^{+/+}$ mice (Fig. 2d,e).

PTEN is considered one of the most prominent tumor suppressors by virtue of its unique PIP$_3$ lipid phosphatase activity, which led us to hypothesize that the lack of tumor predisposition of our C-tail mutant mice might be due to preservation of this key catalytic function despite the low levels at which PTEN$^{S380A}$ and PTEN$^{\Delta 4}$ are expressed. To test this idea, we measured the extent to which PI3K-AKT signaling is inhibited in lung, liver, spleen and prostate of $Pten^{S380A/A}$ and $Pten^{\Delta 4/\Delta 4}$ mice relative to these same four tissues of $Pten^{+/-}$ and $Pten^{+/+}$ mice (Fig. 2f, Supplementary Fig. 3c). We used 2-month-old mice because at this age no tumors are detectable regardless of genotype. None of the $Pten^{+/-}$ tissues analyzed showed evidence of activated PI3K-AKT signaling, as AKT phosphorylation by PDK1 and mTORC2 at residues T308 and S473, respectively, appeared normal. Consistent with this, prominent targets of activated AKT such as TSC2$^{T1462}$, GSK3α$^{S21}$ and GSK3β$^{S9}$, showed normal levels of phosphorylation. By contrast, $Pten^{\Delta 4/\Delta 4}$ mice consistently showed robust phosphorylation at AKT$^{T308}$, TSC2$^{T1462}$, GSK3α$^{S21}$ and GSK3β$^{S9}$ in all four tissues. Increased phosphorylation of these substrates also occurred in prostate and spleen of $Pten^{S380A/A}$ mice, albeit to a lesser extent. Furthermore, phosphorylation at AKT$^{S473}$ was elevated in select tissues $Pten^{\Delta 4/\Delta 4}$ mice. These data indicate that PTEN insufficiency and increased PI3K-AKT signaling per se do not necessarily correlate with tumor predisposition.

### PTEN dose reduction is not a potent driver of prostate cancer

Support that PTEN dosage drives cancer progression originated from the observation that graded reduction of PTEN in mice inversely correlates with prostate cancer progression[7]. This study included $Pten^{+/+}$, $Pten^{+/-}$, $Pten^{-/H}$, and PB-Cre conditional $Pten^{-/-}$ mice, but not $Pten^{H/H}$ (homozygous hypomorphic) mice. This leaves open the possibility that (epi)genetic events that affect the functional integrity of the remaining PTEN produced by prostate epithelial cells of $Pten^{+/-}$ and $Pten^{-/H}$ are the main driver of neoplastic growth. To further investigate this, we wanted to generate a $Pten^{H/H}$ mouse strain with a concordant PTEN reduction to $Pten^{+/-}$ mice, reasoning that if PTEN dosage alone

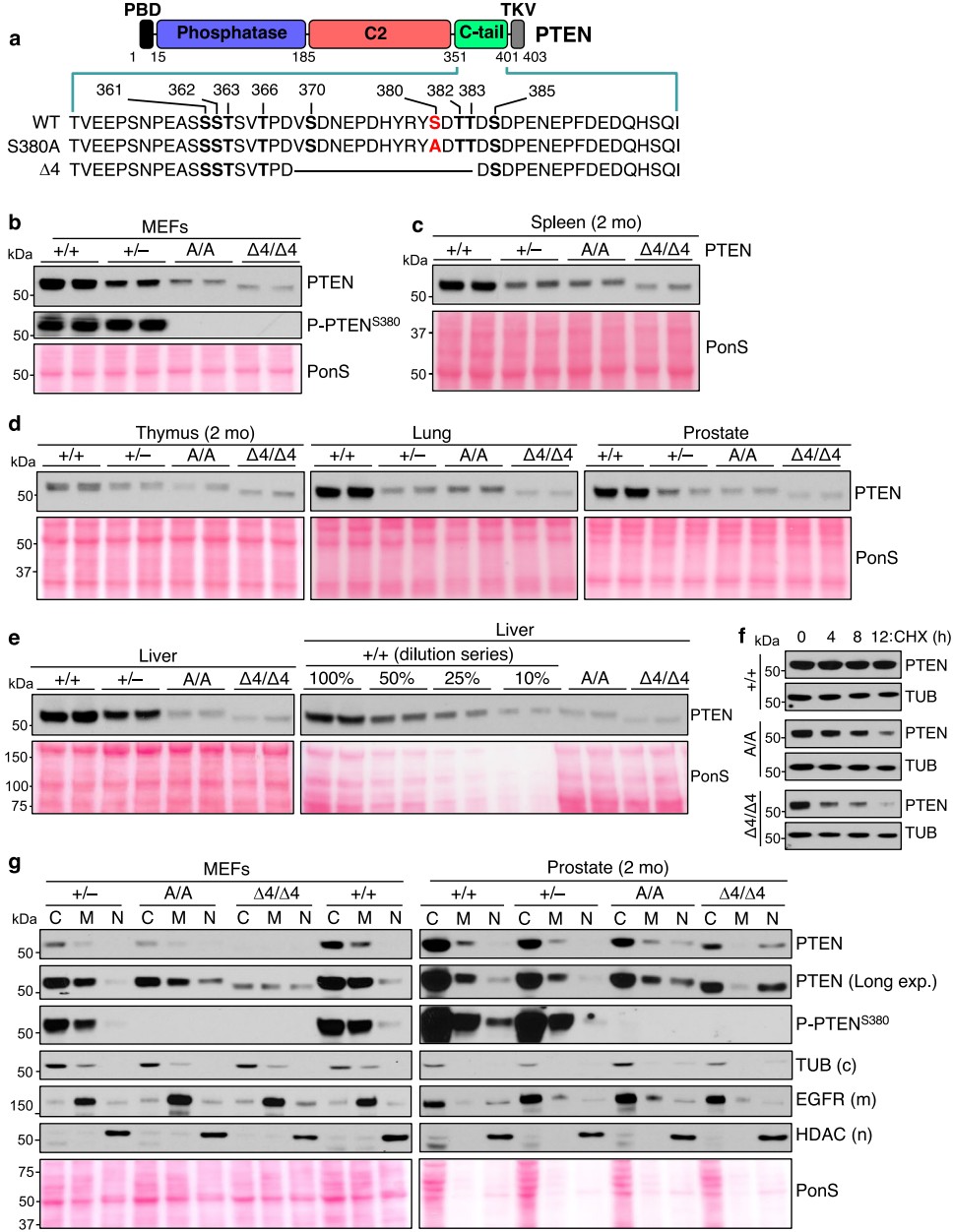

**Fig. 1 | PTEN^{S380A} and PTEN^{Δ4} are unstable but enriched in the nuclear fraction.**
**a** PTEN domain structure highlighting C-tail phosphorylation sites and mutations introduced in *Pten*^{S380A} and *Pten*^{Δ4}. PBD, PIP2-binding domain; TKV, PDZ-binding domain. **b** WB of P5 MEF lysates probed for indicated antibodies. Ponceau S (PonS) staining served as loading control. **c**, **d** WBs of lysates from the indicated tissues harvested at 2 months probed for PTEN. **e** Quantitation of PTEN^{S380A} and PTEN^{Δ4} protein levels in liver using serially diluted liver lysates of +/+ mice as reference. **f** Measurements of PTEN, PTEN^{S380A} and PTEN^{Δ4} protein turnover rates in P4 MEFs by inhibiting protein synthesis with 20 μg/ml cycloheximide for the indicated times.

PTEN levels at each timepoint were assessed by WB for PTEN, with Tubulin as loading control. **g** Cell fractionation of MEFs and 2-month-old prostates of the specified genotypes. Indicated fractions (C, cytoplasm; M, membrane; N, nuclear) were blotted and probed for PTEN, P-PTEN^{S380}, and compartment-specific proteins. Chromatin and cytoskeletal fractions were also analyzed but lacked PTEN and were thus not shown. Blots in (**b**–**e**) are representative of at least 3 individual samples. **f** was performed on 2 independent cell lines and (**g**) on 3 independent prostates. Source data are provided as a Source Data file.

drives tumor formation, both genotypes should theoretically exhibit similar predisposition to prostate tumorigenesis.

To generate a hypomorphic *Pten* allele, we used CRISPR-Cas9 gene editing in fertilized eggs to insert a neo gene cassette in intron 3 of the endogenous *Pten* locus (Fig. 3a). The neo coding sequence is known to contain a cryptic exon that reduces the amount of wildtype transcript and is frequently used to create hypomorphic alleles[39–41]. We used C57BL/6 because this genetic background is frequently used in prostate cancer studies and aligns more closely with earlier PTEN dosage reduction studies, which were performed on a mixed C57BL/6

×129 background[7,8]. Western blot analysis of prostates from mice with two copies of the resulting *Pten* hypomorphic allele (designated HN allele) showed that PTEN levels were reduced compared to *Pten*^{+/−} prostates, but not enough to match levels in prostates of *Pten*^{+/−} males on a C57BL/6 genetic background (Supplementary Fig. 4a,b). To further reduce PTEN dosage we used a second round of editing to insert a polyomavirus polyadenylation (P) sequence that has previously been used to create hypomorphic alleles in mice into the HN allele (Fig. 3a)[39,42]. Western blot analysis of prostate lysates of mice homozygous for the resulting HNP *Pten* allele revealed that their PTEN levels

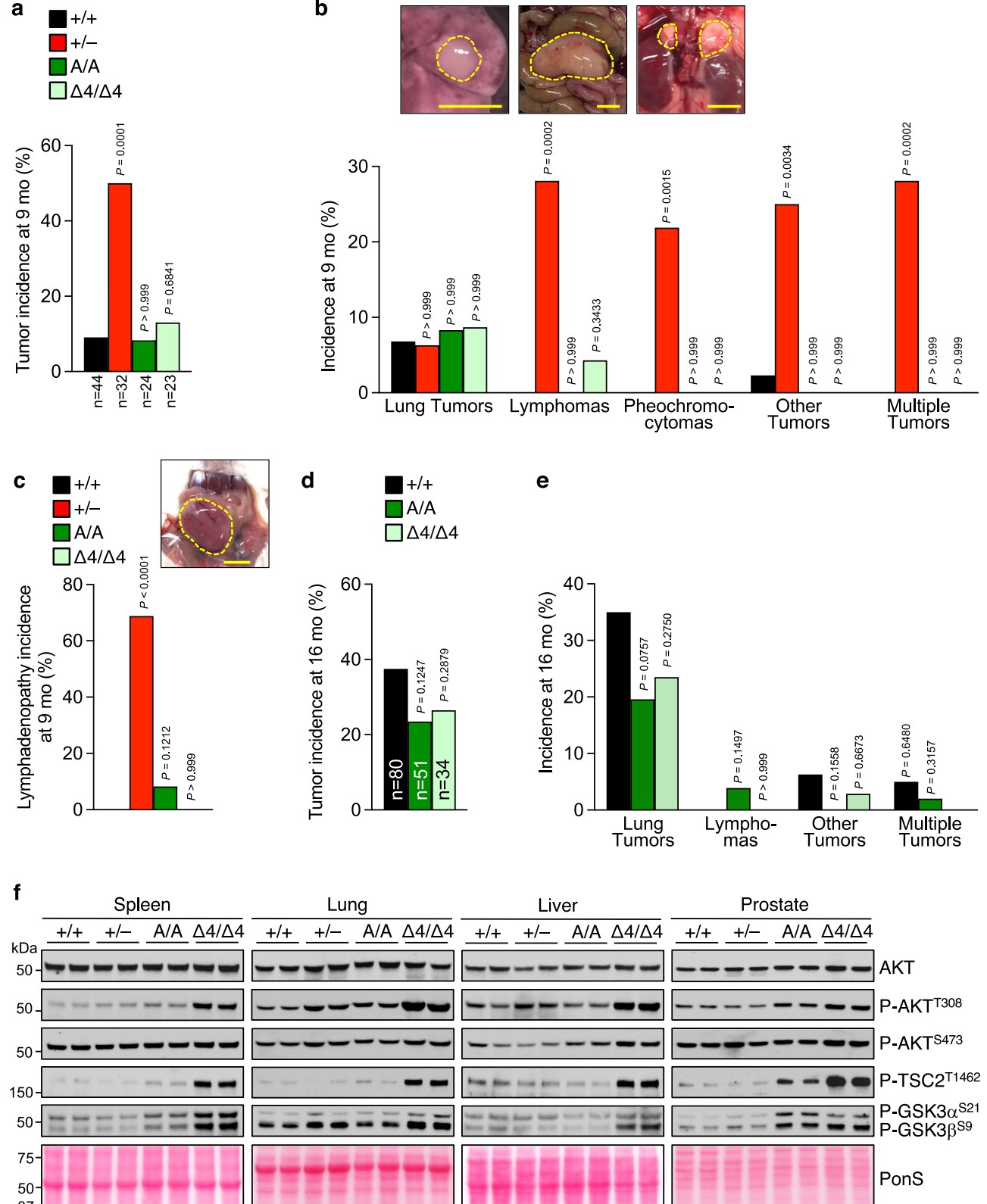

**Fig. 2 | Tumor protection in *Pten*^S380A/A and *Pten*^Δ4/Δ4 mice despite AKT signaling.**
**a** Overall and (**b**) tumor-specific incidence of tumors at 9 months in FVB mice of indicated genotypes. Number of mice (n) for (**a**–**c**) is indicated in (**a**). Incidence of specific tumors is based on macroscopic screening. "Other tumors" include lipomas and tumors in breast, uterus and liver. Photos above graph show examples of below graphed tumors. **c** Incidence at 9 months of lymphadenopathy. We note that the incidence in +/+ was 0%. Photo depicts an example of lymphadenopathy in a sub-mandibular lymph node of a *Pten*^+/− mouse. **d, e** As (**a, b**) but now at 16 months,

number of mice is indicated in (**d**). *Pten*^+/− mice typically don't reach 16 months of age and are therefore excluded. **f** WBs of lysates from indicated tissues and genotypes harvested at 2 months and probed with indicated antibodies. Blot is representative of at least 3 individual samples. Ponceau S (PonS) staining served as loading control. Statistical significance was assessed by two-sided Fisher's exact test compared to +/+. Scale bars are 5 mm. Source data are provided as a Source Data file.

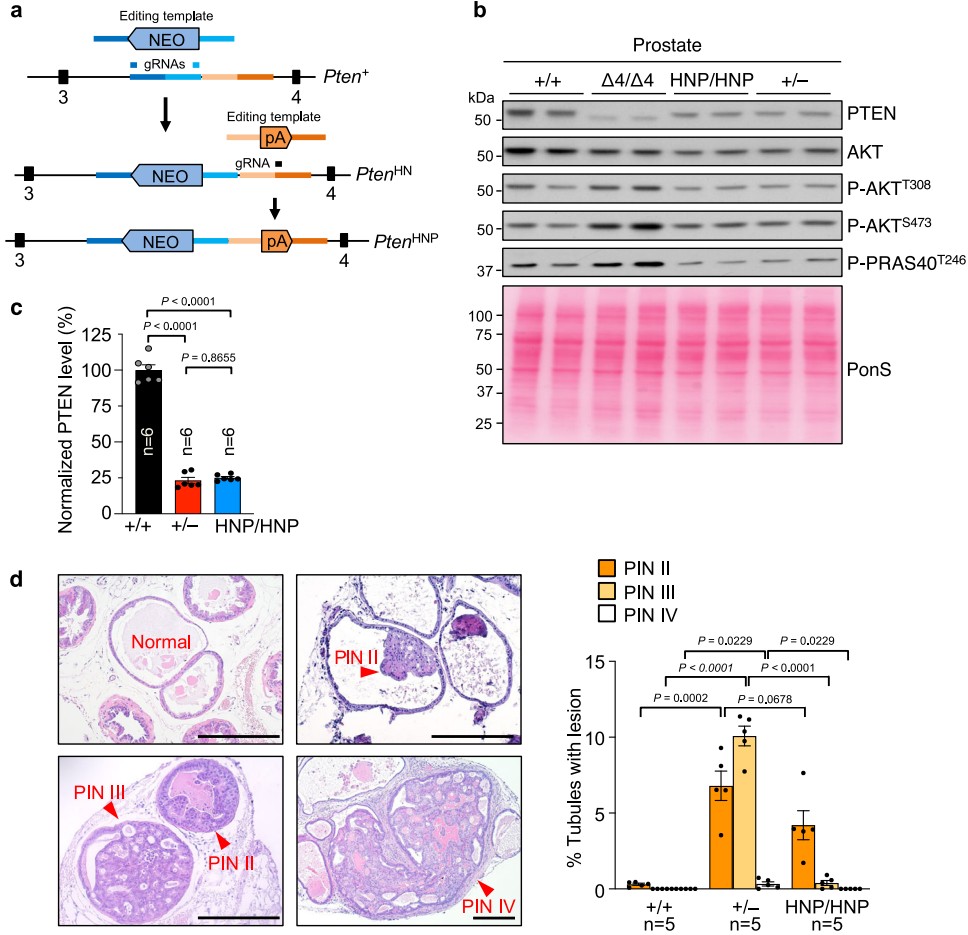

**Fig. 3 | *Pten*^HNP/HNP^ males have a less advanced PIN phenotype than *Pten*^+/−^ males despite similarly reduced PTEN levels. a** CRISPR-CAS9-targeting design of *Pten* hypomorphic mice. *Pten*^HN^, hypomorph Neo; *Pten*^HNP^, hypomorph Neo & PolyA. **b** WB of prostate lysates harvested at 2 months, probed for PTEN and activation of AKT signaling pathway. Blot is representative of at least 3 individual samples. Ponceau S (PonS) staining served as loading control. **c** Quantitation of PTEN level in prostate based on WB. **d** Percentage of prostate tubules with indicated PIN lesions at 6 months. Images depict examples of various PIN lesions. Scale bars are 250 μm. n, number of prostates. Data in (**c** and **d**) are presented as mean values ± SEM. Statistical significance was assessed by ordinary one-way ANOVA, followed by Tukey's multiple comparisons test in (**c**) and Dunnett's multiple comparisons test compared to *Pten*^+/−^ in (**d**). Source data are provided as a Source Data file.

were similarly reduced as in *Pten*^+/−^ prostates (Fig. 3b,c and Supplementary Fig. 4c,d).

Next, we prepared paraffin-embedded prostates of 6-month-old *Pten*^HNP/HNP^ (HNP/HNP), *Pten*^+/−^, and *Pten*^+/+^ mice and serially sectioned these, collecting 5 μm sections for H&E staining and PIN lesion quantitation every 200 μm. Consistent with earlier studies[7,9,19], *Pten*^+/−^ prostates were highly prone to tumorigenesis, with 10% and 0.3% of prostate tubules containing PIN III and PIN IV lesions, respectively. In contrast, *Pten*^HNP/HNP^ prostates had almost no high-grade lesions, showing PIN III lesions in only 0.4% of tubules and no PIN IV lesions. However, PIN II lesions did form at increased rates in *Pten*^HNP/HNP^ prostates compared to *Pten*^+/+^ prostates (4.2% versus 0.3%), indicative of a mild tumor-predisposition (Fig. 3d). The observed differences in lesion incidence and grade give credence to the idea that PTEN dose reduction is not the central driver of prostate tumorigenesis in *Pten*^+/−^ mice.

### PIN lesions formation in *Pten*^+/−^ mice involves further PTEN dysfunction

The most obvious mechanism of PIN lesion formation in *Pten*^+/−^ mice would be a loss of function of the remaining wildtype allele. However, this is difficult to assess by screening for (epi)genetic changes due to the relatively small size of PIN lesions and the presence of non-neoplastic cells. To bypass this problem, we utilized P-AKT^S473^ as a marker for functional loss of PTEN based on the previously reported observation that biallelic inactivation of *Pten* in *PB-Cre;Pten*^Flox/Flox^ mice is characterized by a marked increase in P-AKT^S473^ levels[7]. We independently confirmed this and found high plasma membrane levels of P-AKT^S473^ in *PB-Cre;Pten*^Flox/Flox^ lesions to closely correlate with markedly reduced PTEN immunolabeling (Fig. 4a).

Consistent with P-AKT^S473^ western blot data on prostates of 2-month-old *Pten*^+/−^ mice (Fig. 2f, Supplementary Fig. 3c), normal prostate tissue of 6-month-old *Pten*^+/−^ mice did not show evidence of elevated P-AKT^S473^ immunostaining (Fig. 4a). In contrast, PIN lesions in these prostates consistently exhibited high P-AKT^S473^ and low PTEN labeling, reminiscent of *Pten*^−/−^ PIN lesions (Fig. 4a–c). Strikingly, even emerging lesions showed such staining patterns, suggesting that functional loss of PTEN is an early or initiating event in neoplastic transformation (Fig. 4b, Supplementary Fig. 5a). On the other hand, P-AKT^S473^ and PTEN staining in PIN lesions of 6-month-old *Pten*^HPN/HPN^ mice was indistinguishable from adjacent normal prostate tissue (Fig. 4a,c).

PIN lesions are rare and small in 2-month-old *Pten*^+/−^ mice but continuously increase in size and number as these mice age (Supplementary Fig. 5b). Consistent with this and the above immunolabeling results, complementary western blot analyses of whole prostate lysates of 2-, 6- and 9-month-old *Pten*^+/−^ mice demonstrated a progressive and strong increase in P-AKT^S473^ levels that coincided with

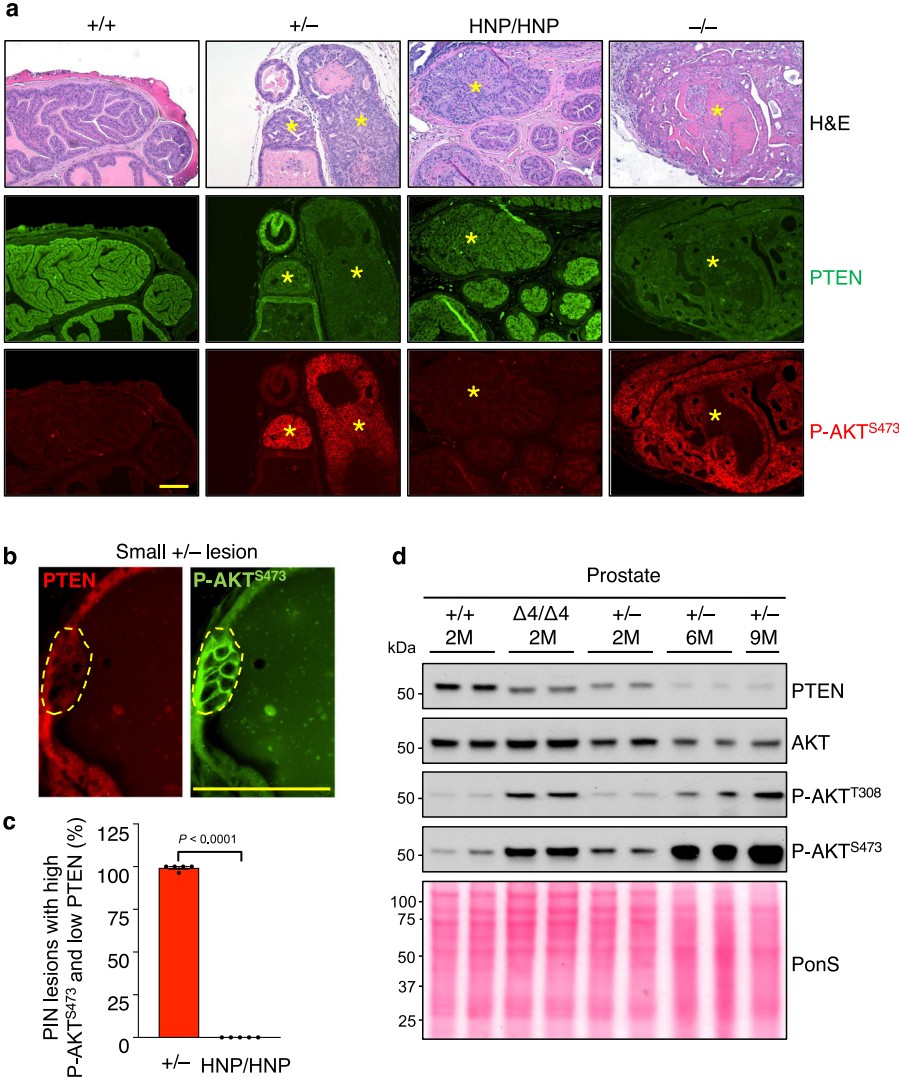

**Fig. 4 | PIN lesion formation in *Pten*+/− males is characterized by loss of PTEN catalytic function. a** Immunostaining of consecutive sections of normal and PIN lesion (*) tubules of indicated genotypes stained for H&E, PTEN and P-AKT^S473. Scale bar is 100 μm. **b** Close-up of a small lesion of epithelial cells that have lost PTEN expression and have gained P-AKT^S473 expression. **c** Quantitation of PIN lesions with increase in P-AKT^S473 expression and simultaneous loss of PTEN. 5 mice per genotype with 12-55 PIN lesions per mouse were analyzed by IF staining. Data are presented as mean values ± SEM. Statistical significance was assessed by two-tailed unpaired *t*-test. **d** WB of lysates from prostates of indicated genotype and age in months (M), probed for PTEN, AKT, P-AKT^T308 and P-AKT^S473. Ponceau S (PonS) staining served as loading control. *N* = 2 individual prostates per genotype and timepoint, except the 9-month sample (*n* = 1). Source data are provided as a Source Data file.

decreasing levels of PTEN (Fig. 4d, Supplementary Fig. 5e,f). Furthermore, comparison to prostate lysates of *Pten*^Δ4/Δ4 mice indicated that the increase in P-AKT in the context of neoplastic growth in *Pten*+/− mice is considerably more robust.

## PTEN^S380D is unstable and drives PIN lesion formation

The observation that *Pten*^S380A/A mice are not tumor prone despite PTEN instability, AKT hyperactivity, and chromosomal instability led us to hypothesize that PTEN molecules that are not phosphorylated at S380 have particularly potent tumor suppressive activity. We further explored this idea specifically within the context of prostate cancer through comparative analysis of PTEN^S380A and its phosphomimetic counterpart PTEN^S380D, which were generated by gene editing C57BL/6 fertilized eggs. Like *Pten*^S380A/A mice, *Pten*^S380D/D (D/D) mice were viable and overtly indistinguishable from *Pten*+/+ littermates.

Western blot analysis of prostate lysates from 2-month-old mice revealed that, just like PTEN^S380A, PTEN^S380D expression was reduced compared to wildtype PTEN (Fig. 5a and Supplementary Fig. 6a).

Homozygous mutation of S380 to A or D had no apparent impact on the phosphorylation status of other C-tail residues as determined by probing western blots with phospho-specific antibodies against S380/T382/T383 and S385 (Fig. 5a, Supplementary Fig. 6b,c). PTEN^S380D distinguished itself from PTEN^S380A with regards to subcellular distribution in that its nuclear levels were reduced rather than preserved as observed for PTEN^S380A (Fig. 5b, Supplementary Fig. 6f).

Cycloheximide chase experiments in MEFs showed PTEN^S380D levels were low due to reduced protein stability (Fig. 5c, Supplementary Fig. 6d). Actinomycin D chase experiments indicated that both the amount and the stability of *Pten* transcripts were normal in *Pten*^S380A/A and *Pten*^S380D/D MEFs (Supplementary Fig. 6e). *Pten* transcripts contain alternative translational start sites, which yield two isoforms, PTENα and PTENβ, with 173 and 166 amino-acid N-terminal extensions, respectively[43,44]. Both these proteins are low abundance proteins relative to PTEN, with PTENα localizing to mitochondria and PTENβ accumulating in the nucleus. To determine the impact of the S380A and S380D mutations on PTENα and PTENβ expression, we examined

cell fractionations of $Pten^{S380A/A}$, $Pten^{S380D/D}$ and $Pten^{+/+}$ MEFs by western blot analysis using a PTEN antibody commonly used for PTEN isoform detection. With long exposure, PTENα and PTENβ were detectable in the membrane and nuclear fractions, respectively, regardless of genotype (Fig. 5d, Supplementary Fig. 6f). In contrast to PTEN, PTENα and PTENβ levels were not negatively impacted by the S380A and S380D mutations and even increased in nuclear fractions of $Pten^{S380A/A}$ MEFs.

In analyzing the impact of the S380D mutation on AKT activity, we found elevated phosphorylation of AKT at T308 and consistent with this, several AKT substrates (Fig. 5e, Supplementary Fig. 7a–d), which is like what we observed in S380A. Furthermore, key mitotic defects of $Pten^{S380A/A}$ MEFs were also observed in $Pten^{S380D/D}$ MEFs, suggesting that PTEN instability rather than lack of S380 phosphorylation affects its centrosome-associated mitotic functions (Supplementary Fig 8).

Next, we quantified PIN lesions in 6-month-old $Pten^{S380A/A}$ and $Pten^{S380D/D}$ mice. PIN lesions were very rare and always early stage in $Pten^{S380A/A}$ mice, just like in $Pten^{+/+}$ mice (Fig. 5f), which is consistent with the generalized lack of tumor predisposition of $Pten^{S380A/A}$ mice on an FVB genetic background. The same was true for $Pten^{\Delta 4/\Delta 4}$ mice (Supplementary Fig. 7e). In contrast, $Pten^{S380D/D}$ mice were prone to prostate neoplasia, with 2% and 0.1% of tubules showing PIN II and PIN III lesions, respectively (compared to 0.2% PIN II and 0% PIN III in $Pten^{+/+}$ mice). At 9 months of age, $Pten^{S380D/D}$ prostates contained 10-fold more PIN III lesions than at 6 months, indicating that the tumor phenotype was progressive (Supplementary Fig. 6g). $Pten^{S380D/D}$ lesions showed no evidence for PTEN loss when analyzed for PTEN and P-AKT$^{S473}$ levels by immunofluorescence (Fig. 5g). These findings, together with similar observations in $Pten^{HNP/HNP}$ mice, indicate that mild prostate tumor phenotypes can develop with certain PTEN insufficiencies but that complete functional loss of PTEN is required for progression to more advanced stages of neoplastic transformation. If so, $Pten^{S380A/-}$ and $Pten^{S380D/-}$ mice would be expected to develop robust prostate cancer phenotypes reminiscent of $Pten^{+/-}$ mice. Indeed, 6-month-old $Pten^{S380A/-}$ and $Pten^{S380D/-}$ mice had similar amounts of PIN II, PIN III, and PIN IV lesions as $Pten^{+/-}$ mice (Figs. 3d, 5g). Importantly, these lesions were characterized by high P-AKT$^{S473}$ and low PTEN compared to flanking normal prostate tissue (Fig. 5h).

## WNT signaling is elevated in $Pten^{S380D}$ prostate epithelial cells

To determine the PTEN tumor suppressive function that requires non-phosphorylated S380, we performed genome-wide transcriptome profiling on prostates of 2-month-old $Pten^{+/+}$, $Pten^{S380A/A}$, and $Pten^{S380D/D}$ mice. At this age, lesions are very rare even in $Pten^{+/-}$ mice, allowing for detection of early transcriptional changes associated with S380 phosphorylation status without transcriptional alterations from neoplastic tissue. $Pten^{S380A/A}$ prostates contained very few differentially expressed genes (DEGs) when compared to $Pten^{+/+}$ prostates (Fig. 6a). In contrast, $Pten^{S380D/D}$ prostates contained 790 DEGs, of which 205 were upregulated and 585 downregulated. Unbiased gene set enrichment analysis (GSEA) indicated that $Pten^{S380D/D}$ prostates might be subject to aberrant signaling through WNT (Fig. 6b), which has been tightly associated with cancer, including prostate cancer[45]. Overrepresentation analyses for gene sets and transcription factor targets of the canonical, the noncanonical $Ca^{2+}$, and the noncanonical PCP WNT signaling pathways indicated that canonical WNT signaling was hyperactive in $Pten^{S380D/D}$ prostates (Fig. 6c,d).

To validate this, we immunostained prostate sections of $Pten^{+/+}$, $Pten^{S380A/A}$, and $Pten^{S380D/D}$ mice for β-catenin, and quantitated epithelial cells with nuclear localization of this protein as a measure for WNT signaling activity. Indeed, $Pten^{S380D/D}$ mice showed evidence of markedly increased WNT activation, with 51% of epithelial cells exhibiting nuclear β-catenin versus 22% and 23% in $Pten^{S380A/A}$ and $Pten^{+/+}$ mice, respectively (Fig. 6e). Furthermore, nuclear β-catenin fluorescence was consistently more intense in prostate epithelial cells of $Pten^{S380D/D}$ than in those of the other two genotypes. Complementary analysis of $Pten^{+/-}$

prostates further indicated that this increase was specific for $Pten^{S380D/D}$ prostates. We also measured Wnt/β-catenin signaling activity using a transgenic reporter that expresses a H2B-EGFP fusion protein under the control of six copies of a TCF/LEF response element and a heat shock protein 1B minimal promoter[46]. Again, $Pten^{S380D/D}$ prostates showed a more than two-fold increase in the proportion of prostate epithelial cells with active WNT signaling compared to $Pten^{+/+}$, $Pten^{+/-}$, and $Pten^{S380A/A}$ mice (Fig. 6f).

## PTEN$^{S380D}$ targets β-catenin to exert oncogenic properties

Expression of non-degradable mutant β-catenin in mice is sufficient to drive tumor formation in multiple tissues, including the prostate[47–49]. These findings, together with data from studies in cancer cell lines showing that PTEN interacts with β-catenin and inhibits its nuclear accumulation and transcriptional activity[50,51], led us to speculate that increased nuclear β-catenin in prostate epithelium of $Pten^{S380D/D}$ mice is due to aberrant PTEN-β-catenin interaction and a driver of PIN lesion formation. Our observation that human recombinant PTEN precipitated human recombinant β-catenin in vitro indicative of direct protein-protein interaction further supported this idea (Fig. 7a). As a first step to investigate this further, we examined PTEN-β-catenin complex formation in the membrane, cytoplasmic, and nuclear fractions of $Pten^{+/+}$, $Pten^{S380A/A}$ and $Pten^{S380D/D}$ prostates using a co-immunoprecipitation approach. PTEN precipitates contained β-catenin in all three fractions regardless of genotype (Fig. 7b). The amounts of β-catenin that precipitated from the membrane or cytoplasmic fractions were comparable across genotypes. By contrast, PTEN$^{S380D}$ consistently precipitated more β-catenin from nuclear fractions than PTEN$^{S380A}$ or wildtype PTEN. In case of wildtype PTEN this is particularly striking because nuclear PTEN$^{S380D}$ levels are considerably lower than those of wildtype PTEN (Fig. 5b).

Next, we examined whether β-catenin hyperactivity underlies PIN lesion formation in $Pten^{S380D/D}$ mice by breeding a $Ctnnb1$ knockout allele into the strain and quantitating PIN lesions at 6 months of age. In this analysis, PIN lesions were rarely observed in $Pten^{S380D/D};Cttnb1^{+/-}$ mice and always early stage, whereas PIN lesion predisposition in $Pten^{S380D/D}$ littermates was consistent with earlier assessments (Fig. 7c). Decreased PIN lesion formation in $Pten^{S380D/D};Cttnb1^{+/-}$ mice coincided with fewer cells exhibiting nuclear β-catenin by immunofluorescence (Fig. 7d). No increase in nuclear β-catenin localization was observed in PIN lesions of $Pten^{+/-}$ mice relative to normal flanking tissue (Fig. 7e). Collectively, these data suggest that PTEN C-tail hyperphosphorylation increases nuclear accumulation of β-catenin to promote neoplastic growth in the prostate.

To explore the potential clinical relevance of this pro-tumorigenic mechanism, we focused on human gastric cancers characterized by hyperphosphorylation of PTEN S380 using RNA-sequencing data available from TCGA[27–29]. Low expression of the PTEN C-tail phosphorylation inhibitor PDZK1 has been reported to correlate with S380 hyperphosphorylation and poor clinical prognosis in gastric cancer[28]. We therefore stratified TGCA gastric tumors based on $PDZK1$ mRNA levels into $PDZK1$ low (bottom 15%) and $PDZK1$ high (top 15%) cohorts, compared these two groups for differentially expressed genes, and then performed an overrepresentation analysis for target genes of LEF1, a β-catenin-dependent transcription factor associated with both gastric and prostate cancer progression[52,53]. Indeed, low $PDZK1$ correlated with LEF1 hyperactivity, whereas high $PDZK1$ did not (Fig. 8a), supporting the idea that PTEN C-tail hyperphosphorylation drives pro-tumorigenic WNT signaling in gastric cancers in which $PDZK1$ expression is compromised. Next, we performed the same analysis for prostate cancers in the TCGA data base. Indeed, low $PDZK1$ expression also correlated with LEF1 hyperactivity in this tumor type (Fig. 8b). Analysis of $PTEN$ transcript levels indicated that gastric and prostate cancers with low $PDZK1$ expression do express $PTEN$ (Supplementary Fig. 9).

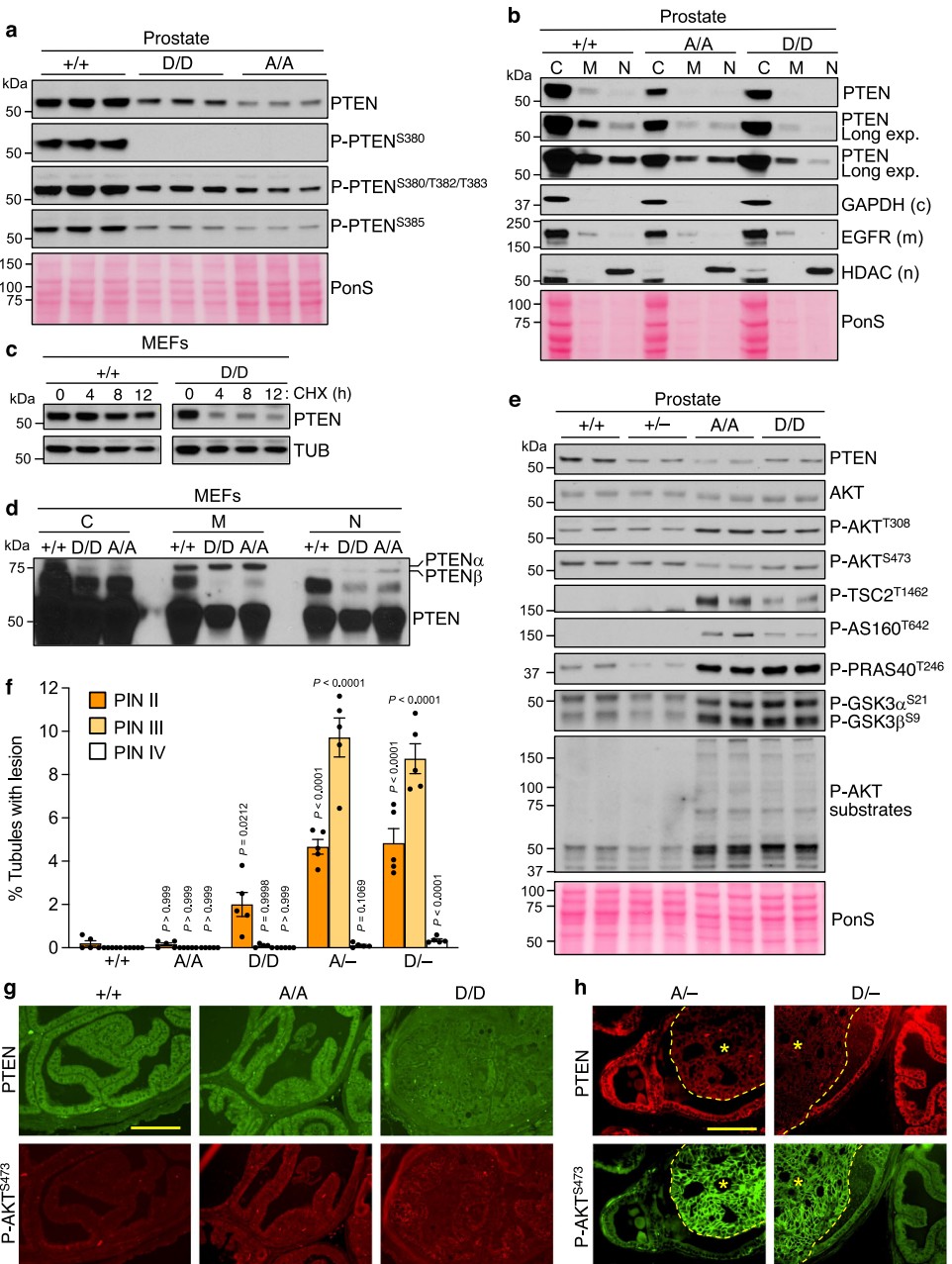

**Fig. 5 | Low PTEN^S380D level causes AKT hyperactivity and PIN lesion formation.**
**a** WB of lysates from 2-month-old prostates (*n* = 3 individual samples) of indicated genotypes. Blots are probed for PTEN and indicated phosphorylated PTEN. Ponceau S (PonS) staining served as loading control. **b** Tissue fractionation of prostates of the specified genotypes. Indicated fractions (C, cytoplasm; M, membrane; N, nuclear) were blotted and probed for PTEN (various exposures) and compartment-specific proteins. *n* = 3 independent fractionation experiments. **c** Measurements of PTEN and PTEN^S380D protein turnover rates in P4 MEFs by inhibiting protein synthesis with 20 μg/ml cycloheximide for the indicated times. PTEN levels at each timepoint were assessed by WB for PTEN, with Tubulin as loading control. *N* = 3 independent experiments. **d** Cell fractionation of MEFs of the indicated genotypes and fractions as in (**b**), blotted and probed for PTEN. Blots were overexposed to be able to visualize PTENα and PTENβ, *n* = 2 independent fractionation experiments.

**e** As in (**a**), but now blots are probed for PTEN, AKT and AKT substrates, and are representative of at least 3 individual samples. **f** Percentage of prostate tubules with indicated PIN lesions at 6 months. A/−, *Pten*^S380A/−; D/−, *Pten*^S380D/−. *N* = 5 mice per genotype. Data are presented as mean values ± SEM. Statistical significance was assessed by ordinary one-way ANOVA, followed by Dunnett's multiple comparisons test compared to +/+. **g** Immunostaining of consecutive sections of prostate tissue of indicated genotypes stained for PTEN and P-AKT^S473, *n* = 5 prostates per genotype. **h** Consecutive immunostaining of prostate tissue of indicated genotypes stained for PTEN and P-AKT^S473-Alexa Fluor 488 conjugate. *N* = 2 prostates per genotype, both stained at two different levels in two independent experiments. *Indicates lesions. Scale bars in (**g** and **h**) are 100 μm. Source data are provided as a Source Data file.

## Discussion

The PTEN C-terminal tail is an intriguing domain in that it is rarely mutated in human cancers (https://cancer.sanger.ac.uk/cosmic), albeit, according to in vitro studies, implicated in the regulation of key PTEN properties linked to tumor suppression, including PTEN stability, localization, phosphatase activity, and homo- and heterotypic protein interactions[12,14,31,33,36–38]. By introducing subtle C-tail domain mutations compatible with mouse embryonic and postnatal development when in homozygosity, we obtained several important insights into PTEN function. First, we demonstrate that *Pten* mutations that substantially

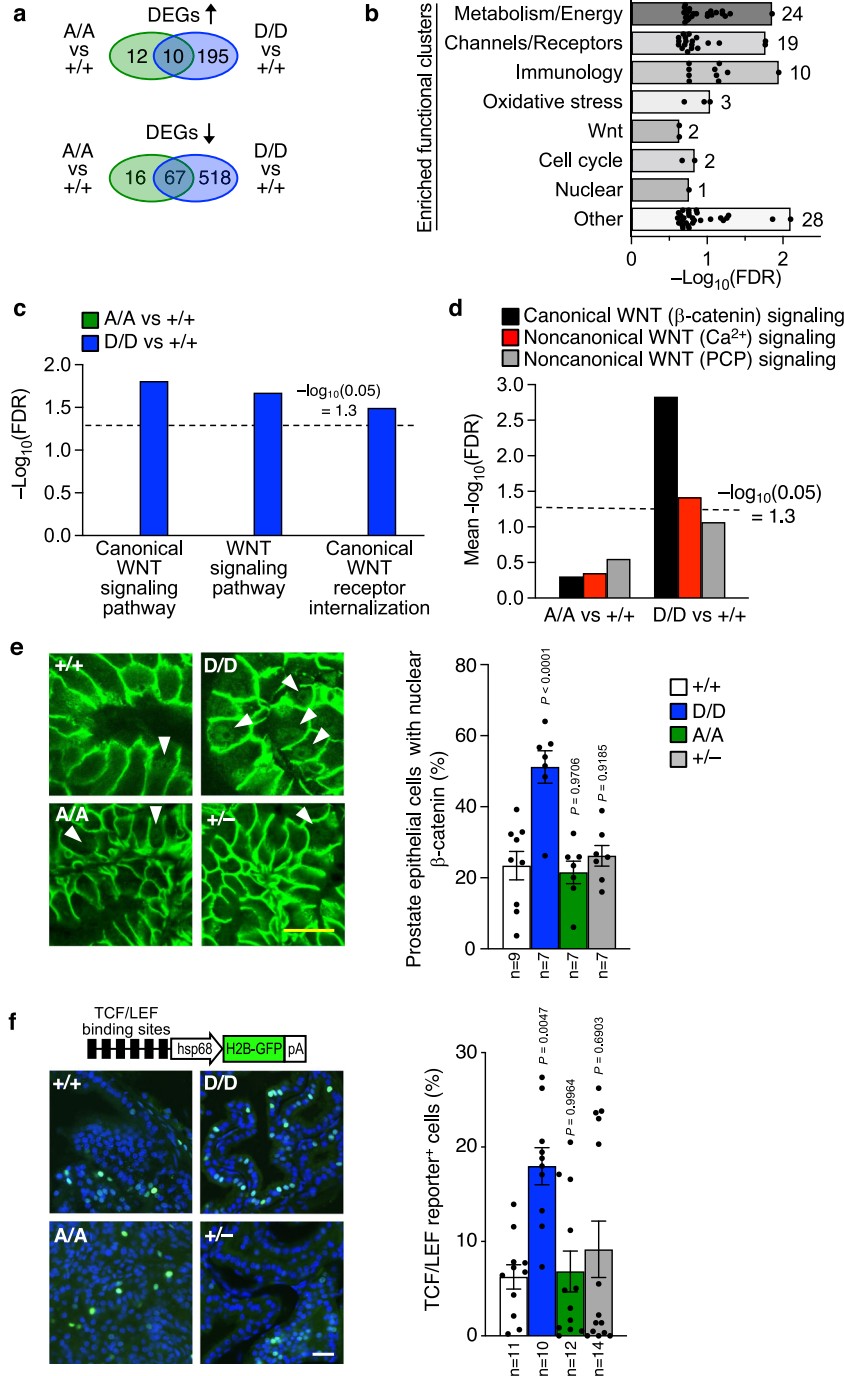

**Fig. 6 | WNT hyperactivity in *Pten*^S380D prostates. a** Venn diagram of differentially expressed genes (DEGs) from RNAseq analyses of prostates from 2-month-old mice of indicated genotypes. ↑, upregulated genes; ↓, downregulated genes. $n = 3$ independent mice/genotype. **b** Annotations from unbiased gene set enrichment analysis. Biological processes clustered by common function. FDR, false discovery rate. Significantly enriched (FDR < 0.25, $-\log_{10} > 0.6$) processes are shown. Bars represent maximum $-\log_{10}$(FDR) per functional group, and dots represent individual annotations for pathways under a given functional group. Numbers next to bars represent the total number of significant biological processes per group. **c** Overrepresentation of gene sets belonging to indicated WNT signaling pathways. **d** Overrepresentation of transcription factor targets belonging to indicated WNT signaling pathways. $-\log_{10} > 1.3$ (FDR < 0.05) in overrepresentation analyses is considered significant. **e** Immunostaining of prostates of indicated genotype stained for total ß-catenin. Arrowheads point to nuclear staining. The percentage of epithelial cells with nuclear staining is graphed. N, number of prostates analyzed, with 126-558 cells counted per prostate. **f** Schematic of TCF/LEF reporter mouse, bred onto indicated genetic background, with representative images of anterior prostate. DNA is marked with Hoechst (blue). Scale bar is 20 μm. The percentage of GFP-positive epithelial cells in anterior prostate is graphed. Data in (**e** and **f**) is presented as mean values ± SEM and statistical significance was assessed by ordinary one-way ANOVA followed by Dunnett's multiple comparisons test compared to +/+. Source data are provided as a Source Data file.

reduce protein level and cause AKT hyperactivity do not necessarily predispose mice to spontaneous tumors. Second, we provide evidence to suggest that initiation of neoplastic growth in prostates of *Pten* haploinsufficient mice requires additional loss of PTEN function. Third, we provide data to suggest that a mutation that mimics hyperphosphorylation of PTEN C-tail residue S380 drives prostate neoplasia in mice through oncogenic activation of WNT signaling and that this mechanism is potentially relevant to human cancers.

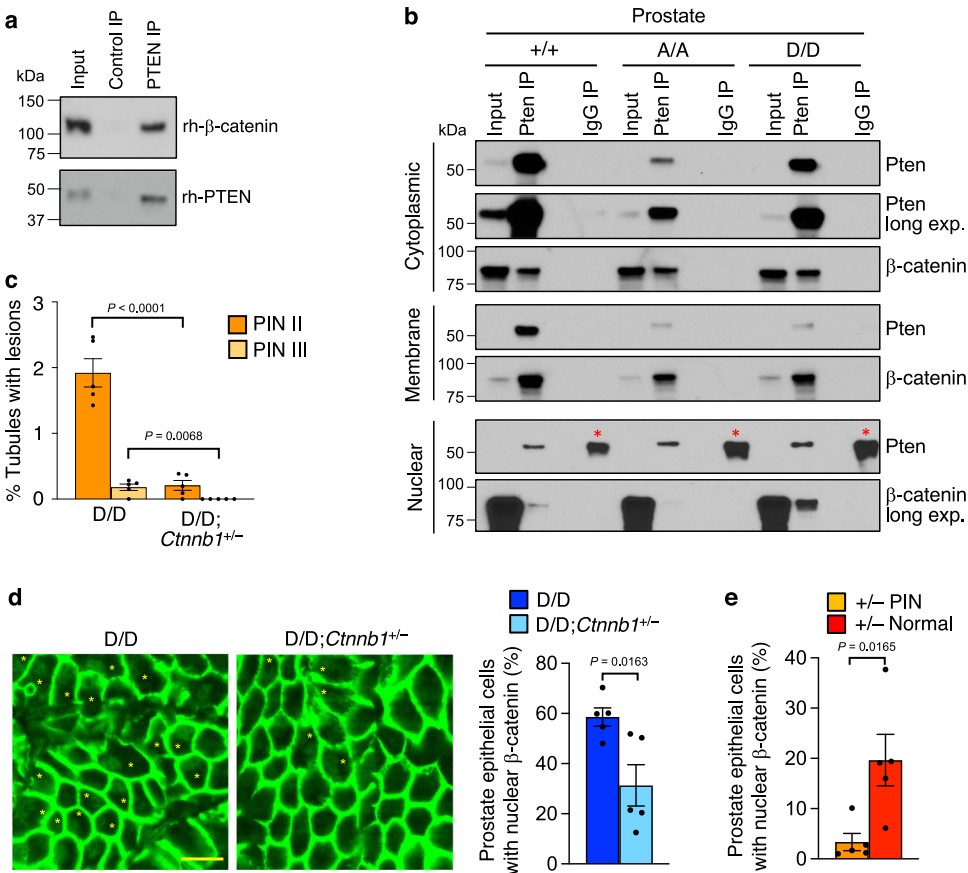

**Fig. 7 | Hyperactive WNT signaling predisposes _Pten_<sup>S380D</sup> mice to PIN lesions.**
**a** Mixtures of recombinant human PTEN and β−catenin proteins subjected to immunoprecipitation with PTEN or corresponding control (anti-RFP) antibodies and analyzed by immunoblotting using PTEN and β−catenin antibodies. The blot is representative of 3 independent experiments. **b** Immunoblot of 2-month-old fractionated prostate lysates of indicated genotypes subjected to immunoprecipitation with PTEN or control antibody and analyzed with the indicated antibodies. Blots represent 3 independent experiments. * Marks IgG band, only visible in the nuclear fraction because of the necessary long exposure to be able to detect the PTEN band.

**c** Percentage of prostate tubules ($n = 5$) with indicated PIN lesions at 6 months. PIN IV were not present and are therefore not depicted. **d** Immunostaining of prostates of indicated genotypes stained for total ß-catenin. Asterisks indicate epithelial cells with nuclear staining. Scale bar represents 10 μm. The percentage of epithelial cells with nuclear staining is graphed. 5 prostates per genotype were analyzed, with 137-230 cells counted per prostate. **e** As in (**d**) but now for _Pten_<sup>+/−</sup> PIN lesions and for _Pten_<sup>+/−</sup> normal surrounding tissue. Data in (**c–e**) are presented as mean values ± SEM. Statistical significance in (**c–e**) was assessed by a two-tailed unpaired _t_-test. Source data are provided as a Source Data file.

A large body of in vitro evidence indicates that C-tail phosphorylation reduces PTEN phosphatase activity[14,27–36]. While data from our in vivo analysis of PTEN<sup>S380D</sup> prostates are consistent with this, those of PTEN<sup>Δ4</sup> and PTEN<sup>S380A</sup> prostates are not, in that both mutants also exhibit reduced phosphatase activity despite a lack of critical phosphorylation. Our assessment of phosphatase activity is based on steady state levels of AKT T308 phosphorylation, which is a direct target of PI3K whose activity PTEN counteracts. AKT hyperactivity was further validated by increased phosphorylation of well-established AKT substrates. Increased AKT phosphorylation, which in PTEN<sup>Δ4</sup> tissues was widespread, did not result in predisposition to tumorigenesis. This is even more surprising given that hitherto PTEN mutations that have been modeled in mice that show evidence of increased AKT phosphorylation are also tumor prone[7,8,21,23]. Strikingly, these models typically show aberrant AKT phosphorylation at the S473 mTORC2 target site. S473 was not subject to hyperphosphorylation in all PTEN<sup>S380A</sup> tissues examined, as well as in a subset of PTEN<sup>Δ4</sup> tissues, indicating that PTEN's ability to inhibit mTORC2 activity is largely preserved[54]. It is therefore conceivable that complementary phosphorylation by mTORC2 at S473 is a requirement to generate oncogenic AKT kinase activity. However, both T308 and S473 are hyperphosphorylated in PTEN<sup>Δ4</sup> prostates without causing tumor predisposition, which would argue against this explanation. It remains

possible that T308 and S473 hyperphosphorylation in this tissue does not reach a level high enough to elicit AKT-mediated oncogenicity.

Beyond insufficiently high T308 and S473 modification, what might be other potential reasons for the uncoupling of AKT hyperactivity and tumorigenesis in our C-tail models? Although membrane-associated and cytoplasmic levels of PTEN<sup>S380A</sup> and PTEN<sup>Δ4</sup> were markedly reduced, nuclear levels of these proteins were not. This raises the possibility that tumor suppressive functions beyond inhibition of AKT phosphorylation remain intact and their perturbation is a requirement for initiation of cellular transformation. The observation that nuclear PTEN<sup>S380D</sup> levels are low supports this idea. It is conceivable that PTEN that cannot be phosphorylated at S380, which applies to both PTEN<sup>S380A</sup> and PTEN<sup>Δ4</sup>, reinforces a prominent tumor suppressive function that counteracts the pro-tumorigenic effects of uncontrolled PI3K-AKT signaling. This does not include PTEN's role in faithful chromosome segregation because all C-tail mutants exhibited centrosome-associated mitotic defects and were prone to aneuploidy. The abovementioned subcellular localization data on PTEN S380A and PTEN S380D highlight their distinct cellular properties and are in keeping with previously reported data indicating that the S380A mutation drives nuclear import by exposing the N-terminal PTEN NLS motif[55]. The distinct properties of PTEN S380A and PTEN S380D are further underscored by our observation that nuclear β-catenin

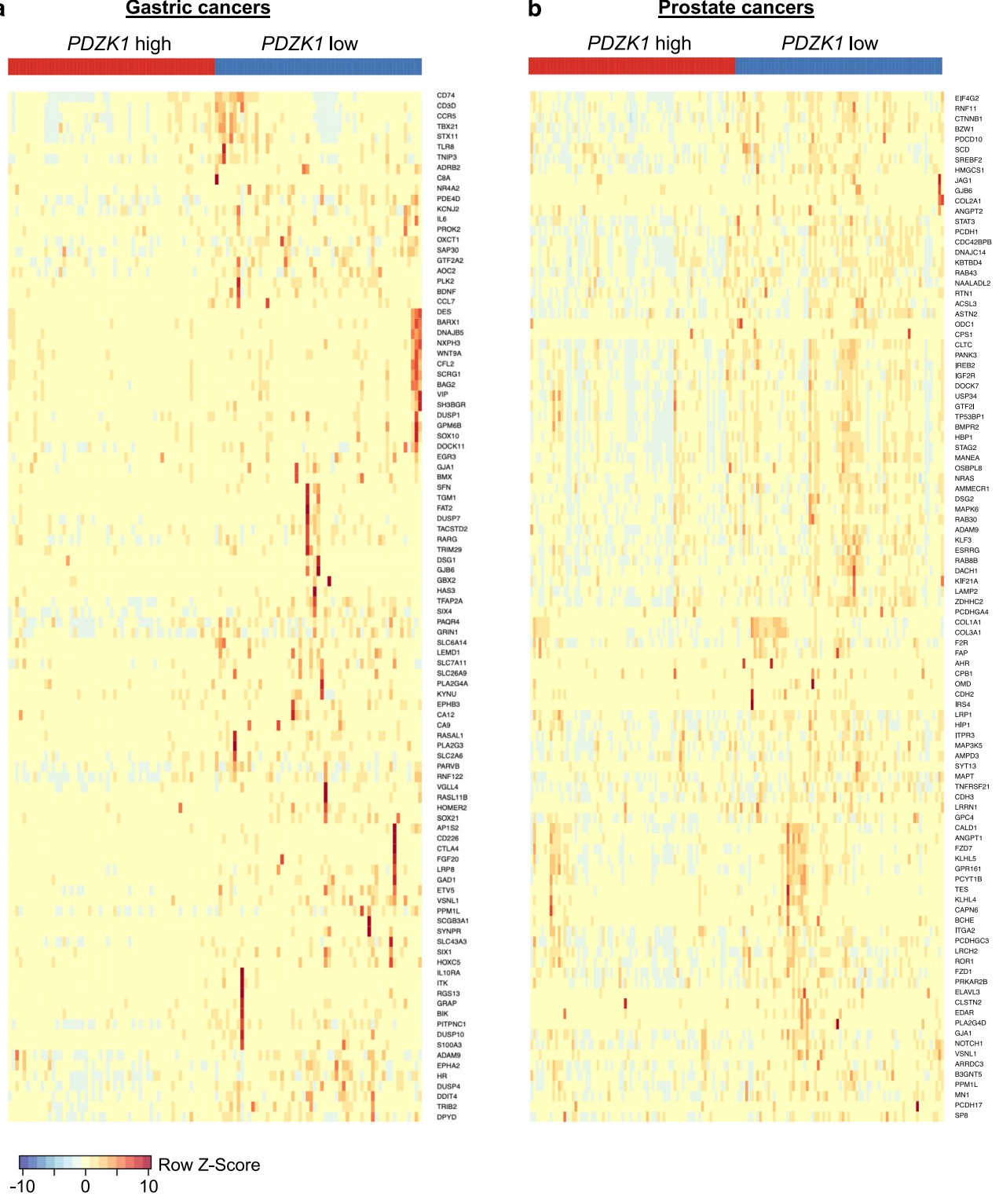

**Fig. 8 | Low *PDZK1* expression in human gastric and prostate cancers correlates with increased *LEF1* activity. a** Heat map (Row *Z*-Score) of the 100 top differentially expressed LEF1 target genes with FDR < 0.05 in TGCA-STAD gastric cancers with low *PDZK1* expression (bottom 15%; *n* = 57 cases) versus high *PDZK1* expression (top 15%; *n* = 57 cases). **b** As in (**a**) but now for TGCA-PRAD prostate cancers with *n* = 75 cases in the bottom 15% and *n* = 75 cases in the top 15% of *PDZK1* expression.

interacts more abundantly with PTEN S380D than with PTEN S380A, even though nuclear PTEN S380D levels are substantially lower.

Another feature of C-tail phosphorylation in vitro, PTEN protein stabilization, was observed in vivo, in that both PTEN^S380A and PTEN^Δ4 were expressed at markedly reduced levels. PTEN^S380D, however, was also expressed at subnormal levels which at the surface seems

inconsistent with what has been previously reported on PTEN C-tail phosphorylation. However, these earlier studies only tested PTEN stability of combination D mutants of S380/T382/383 (D3) or S380/T382/383/S385 (D4) and did not include D mutants for the individual residues. Furthermore, it is important to consider that our experimentation was conducted in vivo, under physiological conditions with

endogenous mutants, whereas the earlier work on C-tail phosphorylation was done in cancer cell lines with ectopically expressed PTEN variants. It should also be noted that even though A and D substitutions are the best available approaches to studying the different phosphorylation states of serine residues, these substitutions carry the risk that they may not accurately model PTEN's actual phosphorylation states. Assuming they are accurate, our observations in our studies of S380 mutant mice suggest that dynamic phosphorylation-dephosphorylation of S380 is critical for PTEN stability and PI3K-AKT inhibition. Unlike PTEN, PTENα and PTENβ do not appear to be unstable further indicating that their post-translational regulation is distinct from canonical PTEN[56,57]. Whether PTEN stability and PI3K-AKT inhibition are similarly regulated by C-tail T382, T383 and S385 remains to be determined as the phosphorylation status of these residues did not seem to be impacted by S380 mutation.

PTEN stability is considered an important requirement for its tumor suppressive ability. On the surface, our results showing that mice with low PTEN levels retain the ability to suppress tumorigenesis seems to argue against this idea. However, a model that distinguishes between low amounts of wildtype PTEN and low amounts of mutant PTEN with or without the presence of wildtype PTEN has the potential to unify all the data. Specifically, because PTEN homodimerizes, the amount of functional complex may be low due to the dominant-negative actions of certain mutants, as was elegantly demonstrated in PTEN strains heterozygous for C124S or G129E[17]. The same may hold true in instances where a wildtype allele is paired with one of the many nonsense mutants found in human cancers. In our experiments, mice express low amounts of mutant PTEN protein in the absence of its wildtype counterpart, which is a different setting then low amounts of wildtype PTEN. Although the mutations we introduced do not mimic mutations found in human cancers, they provide important insights into the mechanisms of PTEN-mediated tumorigenesis that could be explored further as entry points for the development of new therapeutics tailored to a mainstream tumor suppressor. For instance, by interfering with C-tail phosphorylation, it may be possible to potentiate the tumor-suppressive potential of any residual PTEN present in cancer cells, thereby perhaps limiting their ability to survive and proliferate.

Homozygous inactivation of conditional *Pten* knockout alleles with a wide variety of Cre drivers has firmly established that complete loss of PTEN results in neoplastic transformation and robust constitutive PI3K-AKT-mTORC1 signaling[23]. However, *Pten* heterozygous mice often form tumors without evidence of sustaining loss of the wildtype *Pten* allele[8,18,19], which led to the idea that PTEN is a haploinsufficient tumor suppressor, with partial reduction in wildtype PTEN level driving neoplastic transformation. However, a similar reduction in wildtype PTEN caused by homozygosity for a hypomorphic *Pten* allele did not replicate the robust tumor phenotype observed in *Pten* haploinsufficient prostates, which prompted us to further investigate the mechanism of prostate tumorigenesis in *Pten*[+/−] mice. We demonstrate that neoplastic growth in *Pten*[+/−] prostates is consistently associated with robust AKT S473 phosphorylation, which is a characteristic of complete PTEN loss[7]. We find that early-stage lesions consisting of small numbers of cells always exhibit robust AKT S473 phosphorylation. Importantly, such hyperactivity reliably coincides with a decline in PTEN immunofluorescence signal. These data indicate that a second event that compromises the function of the remaining wildtype *Pten* allele is a requirement for initiation of neoplastic growth. The finding that a knockout *Pten* allele in combination with a *Pten*[S380D] allele which affects PTEN stability and has pro-tumorigenic properties seems to require a further loss of PTEN catalytic function to robustly initiate of PIN lesion formation further supports this conclusion.

Two observations argue against a critical tumor suppressive role for the PTEN C-tail domain. The first is our finding here that deletion of a relatively large portion of the C-tail (Δ4) does not promote tumor formation. The second comes from the analysis of COSMIC mutational signatures of human cancers showing that the C-terminal tail domain is rarely a direct mutational target. However, in gastric cancers C-tail hyperphosphorylation correlates with increased AKT phosphorylation, thus presenting a C-tail-related mechanism by which PTEN may lose its tumor suppressive ability[27,28]. Although tumor promoting effects of excessive C-tail phosphorylation could, in principle, also be accomplished by mutation of S380 or other phospho-residues to phosphomimetics, such conversions would each require multiple nucleotide substitutions and would therefore be unlikely. Aberrant kinase activity, on the other hand, is a much more realistic scenario for hyperphosphorylation of PTEN C-tail residues given that the main kinase involved, CK2, is frequently overexpressed or otherwise hyperactivated in human cancers[58]. Although uncontrolled C-tail phosphorylation may drive tumorigenesis through loss of PTEN catalytic activity, our discovery that PTEN[S380D] in mice drives neoplastic growth through hyperactivation of the WNT/β-catenin pathway, implies that excessively phosphorylated PTEN can also act in an oncogenic manner. The concept that tumor suppressors can serve as macromolecular double agents with tumor suppressive and oncogenic functions is not new. For instance, for p53 it has been reported that several tumor-associated p53 mutants lack key tumor-suppressive functions while gaining new activities to promote tumorigenesis[59,60]. In this context, it is important to emphasize that although we find that oncogenic β-catenin/WNT signaling is critically required for PIN formation in Pten[S380D/D] mice, it is well possible that the observed partial loss of tumor suppressive PTEN functions (as reflected in the increase in PI3K-AKT signaling and mitotic errors) cooperates with aberrant β-catenin/WNT signaling.

In probing the potential clinical relevance of this oncogenic PTEN mechanism, we obtained transcriptomic evidence suggesting that WNT/β-catenin activity is indeed elevated in gastric tumors with increased PTEN C-tail phosphorylation due to PDZK1 insufficiency. Interestingly, Heliobacter (H.) pylori infection, the primary known risk factor of gastric cancer, yields malignancies that are characterized by elevated WNT/β-catenin signaling and increased PTEN phosphorylation at S380, T382 and T383[29,61]. This, together with the observation that H. pylori induces phosphorylation at these same residues in cultured gastric epithelial cells[32], raises the possibility that bacterially induced PTEN C-tail phosphorylation drives aberrant WNT signaling during gastric tumorigenesis. The benefits of H. pylori eradication to gastric cancer patients[62,63] may thus in part be attributable to normalization of PTEN C-tail phosphorylation. In extending our transcriptomic analysis to human prostate cancers, we found low *PDZK1* to also correlate with hyperactivation of the WNT/β-catenin pathway in this tumor type. It should be noted that low PDZK1 expression is likely to also affect the phosphorylation status of proteins other than PTEN. Anyways, our experiments set the stage for future immunostaining experiments examining whether PTEN C-tail hyperphosphorylation correlates with nuclear accumulation of β-catenin. For this purpose, we will be using samples of prostate cancer patients that have previously been characterized for PTEN expression status[64]. Also, given the mounting evidence for pro-tumorigenic effects of PTEN C-tail phosphorylation, it will be interesting to further explore the therapeutic relevance of CK2 inhibitors such as CX-4945 (Silmitasertib) for the treatment of gastric cancers and other malignancies characterized by CK2 overexpression and elevated PTEN-C-tail phosphorylation[65].

## Methods

The research presented complies with all relevant ethical regulations. Studies involving laboratory mice were reviewed and approved by the Mayo Clinic Institutional Animal Care and Use Committee, in compliance with national ethical guidelines. The study has been reviewed

by the Mayo Clinic Conflict of Interest Review Board and is being conducted in compliance with Mayo Clinic Conflict of Interest policies.

## Mouse strains

All mice were housed in a pathogen-free barrier environment with ad libitum access to food and water and 12 h light and dark cycles. The temperature was kept between 68 and 79°F, with an average of 71°F, and a humidity target range between 30 and 70%, with an average of 45%. Mice were euthanized by CO2 inhalation. According to Mayo Clinic Institutional Animal Care and Use Committee policy, a mouse with a tumor meeting any of the following criteria requires immediate euthanasia: size being 10% of body weight, ulceration, or interference with vital functions. No animal in this study exceeded or violated these conditions. All non-purchased mouse strains were created by Mayo Clinic's Gene Knockout and Transgenics core facility. Standard gene targeting and embryonic stem cell technology was used to create *Pten*[TKV] mice[26].

*Pten*[S380A] mice were generated by CRISPR-Cas9-mediated gene editing in FVB zygotes (Envigo) by injecting Cas9 mRNA, a gRNA 5′-TCAGAATATCTATAA TGATC-3′ (guide target sequence) and a single-strand oligodeoxynucleotide (ssODN) 5′-GGTTCATTCTCTGGATCAG AGTCAGTGGTGTCGGCATATCTAT AATGATCAGGTTCATGTCACTAA CATCTGGAGTCACAGAAGTTGAACTGCTA GCCT-3′ (Supplementary Fig. 1a). The same Cas9 mRNA, gRNA and ssODN were used to obtain *Pten*[S380A] mice on a C57BL/6 background, but here the injections were in C57BL/6 zygotes. *Pten*[S380D] mice were directly generated in C57BL/6 background with the same gRNA as for *Pten*[S380A] but with the following ssODN, 5′-AAACAGTAGAGGAGC CAT CAAA TCCA GAGGCTAGCAGT TCAACTTCTGTGACTCCAGATGTTAGTGACAATG AACCTGATCATTAT AGATATGACGACACCACTGACTCTGATCCAGAGAATGAACCTTTTGAT GAAGATCAGCATTC ACAAATTACAAAAGTCTGA-3′. For each of these strains we obtained multiple independent founder lines. DNA sequencing confirmed that they contained the anticipated nucleotide changes designed to create the S380 to A or D substitutions. To minimize the risk of off-target genetic alterations introduced with CRISPR/CAS9 mediated mutagenesis that could impact PTEN function beyond the S380D substitution, we backcrossed the initial FVB and C57BL/6 founder mice two times to wildtype FVB and C57BL/6 mice, respectively, before intercrossing them to generate the homozygous animals. To further minimalize the risk for off-target alterations we performed complementary transcript and protein analysis.

The *Pten*[Δ4] allele was a by-product of *Pten*[S380A] allele generation in FVB. To generate *Pten*[Δ4] on a C57BL/6 background, these mice were backcrossed to C57BL/6 at least 8 times. *Pten*[+/−] were created in FVB zygotes by targeting exon 2 with gRNA 5′-TTTCCTGCAGAAAGA CTTGA-3′ and CRISPR-Cas9-mediated gene editing, which resulted in a four-nucleotide deletion (Supplementary Fig. 3a). For FVB-related experiments, these mice were backcrossed to FVB at least twice before experimentation. To establish *Pten*[+/−] on a C57BL/6 background these mice were backcrossed to C57BL/6 at least 8 times. *Pten*[HN] were generated in C57BL/6 using gRNA 5′-GGGATCCC AATTGCCTGCGG-3′ to insert an inverted Neo gene (from pMC1neo polyA, Addgene 213201) into intron 3 of *Pten*, using 940 bp 5′ and 890 bp 3′ homology arms. *Pten*[HNP] used zygotes from *Pten*[HN] to insert a polyoma polyadenylation sequence into intron 3 that already contained the *Neo* gene. To this end, gRNA 5′-TACCGACTAGTCTACCACAG-3′ and ssODN 5′- TTAT TCTGTCTTTTTATTGCCGATCCAGTGGATACTAGAGTAACACAAAAGA TATGGATTTTC TTGTTTTATTTAGTTTTTTAGTTTTTTGAAAACTGAA ATTTTCTATGTACCAA-3′ were used (Fig. 3a).

*Pten*[−/−] prostates (harvested at 6 months) were derived from *Pten*[Flox] mice (Jackson lab #006440) crossed with PbsnCre transgenic mice (Jackson lab #026662) to establish prostate-specific deletion of PTEN. TCF/LEF1 reporter mice (Jackson lab #013752) were crossed onto various *Pten*-mutant backgrounds. β-catenin floxed mice (Jackson lab #022775) were crossed with HPRT-Cre transgenic females to generate *Ctnnb1*[+/−] mice, which were subsequently bred onto a *Pten*[S380D/D] background. Experiments in Figs. 1–2 and Supplementary Figs. 1–3 used mice on an FVB background, while Figs. 3–7 and Supplementary Figs. 4–8 are based on C57BL/6. Mice in the spontaneous tumor susceptibility study were euthanized at either 9 or 16 months of age and subjected to macroscopic screening for overt tumors. Lungs and adrenal glands were screened with low magnification of a stereo microscope (Olympus SZX16). Tumors were harvested and processed for standard histopathology.

## Cell culture

*Pten*[+/−], *Pten*[ΔTKV/ΔTKV], *Pten*[S380A/A], *Pten*[S380D/D], *Pten*[Δ4/Δ4], *Pten*[Flox/Flox] and *Pten*[+/+] MEFs were generated from independent E13.5 embryos and expanded at 3% O2 in DMEM medium supplemented with 10% heat-inactivated fetal bovine serum, l-glutamine, nonessential amino acids, ß-mercaptoethanol, gentamicin and sodium pyruvate. HeLa cells (ATCC, CCL-2) were expanded at 95% air/5% CO2 in the same medium.

## Western blot analysis, cell fractionation, immunoprecipitations and cycloheximide chase

Western blot analysis and immunoprecipitation were carried out according to standard procedures. Tissue and MEF lysates were prepared in NP40 lysis buffer with protease inhibitors, mixed with Laemmli Sample Buffer and boiled for 10 min prior to gel loading. Subcellular fractionation was performed using the protein fractionation kit (78840 and 87790; Thermo-Fisher) according to manufacturer's instructions. In vitro protein binding studies were conducted in 100 μl PBS at RT using recombinant human PTEN (R&D systems Cat. #847-PN) and recombinant human β-catenin (Sino Biological Cat. #11279-H20B). 250 ng PTEN and 500 ng β-catenin were incubated for 60 min after which 1 μg rabbit polyclonal antibodies against C-terminal human PTEN (Invitrogen Cat. #51-2400) or 1 μg Red Fluorescent Protein (control; Invitrogen Cat. #MA5-15257) were added. After 45 min, 20 μl 50% protein G agarose suspension (in PBS) was added and after 45 min the protein G agarose was washed 5 times with 200 μl PBS supplemented with 0.05% Tween 20. Proteins bound to Protein G agarose were then eluted by addition of 20 μl Laemli Sample Buffer and 5-min incubation at 100 °C and analyzed by western blotting using antibodies against β-catenin (1:2000, BD 610153) and PTEN (1:4000, anti-human PTEN [6H2.1] Cascade Cat. #ABM-2052). For cycloheximide chase experiments, MEFs were cultured in presence of 20 μg/ml cycloheximide for 0, 4, 8 or 12 h prior to harvesting the cells to make lysates. Densitometry quantification was performed using ImageJ (v.1.52a) software by measuring the area under the curve for the band intensity in each lane. Antibodies used were as follows: PTEN (1:2000, 9559, Cell Signaling Technology); P-PTEN[S380] (1:1000, 9551, Cell Signaling Technology); P-PTEN[S380/T382/T83] (1:1000, 9549, Cell Signaling Technology), P-PTEN[S385] (1:1000, 07-890-I Sigma-Aldrich), Tubulin (1:2000, 2125, Cell Signaling Technology); EGFR (1:1000, 71655, Cell Signaling Technology); HDAC (1:1000, Ab7028, Abcam); AKT (1:1000, 9272 Cell Signaling Technology); P-AKT[T308] (1:1000, 2965, Cell Signaling Technology); P-AKT[S473] (1:1000, 9271, Cell Signaling Technology); P-TSC2[T1462] (1:500, 3617, Cell Signaling Technology); P-GSK3α[S21]/β[S9] (1:1000, 9331, Cell Signaling Technology); P-PRAS40[T246] (1:2000, 2997, Cell Signaling Technology); P-AS160[T642] (1:500, 8881, Cell Signaling Technology); P-AKT substrates (1:500, 9614, Cell Signaling Technology); GAPDH (1:1000, 3683 S, Cell Signaling Technology); IgG (1 μl/IP, 0107-01 and 0111-01, Southern Biotech); β-catenin (1:2000, 610153, BD). Secondary antibodies were goat anti-mouse HRP-conjugated and goat anti-rabbit HRP-conjugated (1:10,000, 115-035-146 and 111-035-003, Jackson Immunoresearch). Equal loading was confirmed by Ponceau S staining.

## Mitotic analyses

All mitotic analyses have been previously described[26]. Briefly, chromosome counts were performed on metaphase spreads from

colcemid-treated MEFs. Chromosome segregation errors were tabulated by following live MEFs (at least 28 cells per cell line), transduced with lentiviral TSIN-histone 2b-mRFP to mark DNA red, through mitosis via live cell imaging with a microscope system (Axio Observer; (Carl Zeiss) with $CO_2$ Module S, TempModule S, Heating Unit XL S, a plan Apo 63× NA 1.4 oil differential interference contrast III objective (Carl Zeiss), camera (AxioCam MRm; Carl Zeiss), and Zen Blue software (Carl Zeiss). For spindle geometry analysis, γ-tubulin/α-tubulin- stained MEFs in metaphase, with centrosomes in the same focal plane, were imaged using a confocal laser-scanning microscope (LSM 880; Carl Zeiss) on an Axio Observer Z1 inverted microscope with spectral detectors (32ch 2PMT GaAsP; Carl Zeiss) and a water-immersion lens (C-Apochromate 40x/1.2 NA Korr. FCS; Carl Zeiss). ZEN Black v.14.0.24.201 software (Zeiss) was used to measure the angle between the spindle and the metaphase plate (DNA marked with Hoechst). Cells that had an acute angle between the spindle pole axis and the metaphase plate of <85° or more than 95° were considered asymmetrical. For centrosome distance measurements in prophase, cells were stained for pH3S10/γ-tubulin/Hoechst, and images were taken by laser-scanning microscopy as described above of cells with centrosomes in the same focal plane. The distance between centrosomes (γ-tubulin signals) was measured using ZEN Black software (Zeiss). Prophases with centrosome distance/nuclear diameter ratio ≤0.5 were considered delayed.

## Histopathology and immunostaining

For *prostate sectioning and PIN lesion scoring*, whole prostates were typically isolated at 6 months, formalin-fixed and paraffin-embedded with the anterior lobes flattened at the bottom of the block (come across first while sectioning). The entire prostate was then sectioned in 5 μm sections, collecting two sections per slide and 5 slides per level. After each level we skipped 150 μm, before collecting at the following level. The first slide at each level was used for H&E staining and PIN lesion scoring according to Park et al.[66], which meant that we analyzed every 200 μm. Per level we counted the total number of tubules and the total number of each type of PIN. In the end, all levels would be added together, and the percentage of PIN-positive tubules calculated.

For *immunostaining of tissue*, tissues were collected, formalin-fixed, paraffin-embedded and sectioned at 5 μm thickness. According to standard technique, paraffin sections were rehydrated prior to antigen retrieval with low pH citrate buffer (Vector labs) for 20 min in a pressure cooker, followed by treatment with Image-It FX Signal enhancer (Invitrogen), blocking and antibody incubation overnight at 4 °C. Alexa Fluor-conjugated goat-anti-rabbit, goat-anti-mouse or isotype-specific goat-anti-mouse secondary antibodies were used followed by Hoechst staining to mark DNA. In the case of double staining with PTEN and P-AKTS473-conjugated Alexa fluor 488, the slides were first stained for PTEN, washed, incubated with secondary antibody, washed, blocked and then incubated with the conjugated P-AKTS473 antibody. In the case of the TCF/LEF reporter mice, tissues were fixed in 4% PFA for 2 h, followed by 30% sucrose in PBS overnight before being snap frozen in OCT in cryomolds. Cryosections were cut and DNA was marked with Hoechst staining, while cells expressing the reporter gene were marked by endogenous GFP.

For *immunostaining of MEFs*, in case of Eg5, γ- and α-tubulin staining, cells were fixed in PBS/1% paraformaldehyde (PFA) for 5 min at RT, followed by ice-cold methanol for 10 min and permeabilization with 0.2% Triton X-100 for 10 min at RT. For PTEN staining, MEFs were fixed in 3% PFA for 12 min at RT, followed by permeabilization with 0.2% Triton X-100 for 10 min at RT.

*Primary antibodies used* were as follows: Eg5 (1:100, TA301478, Origene); α-tubulin (1:1,000, T9026 or 2125 S, Sigma); γ-tubulin (1:300, T6557 or T5192, Sigma); P-AKTS473 (1:100, 4060, Cell signaling Technology); P- AKTS473-Alexa Fluor 488 conjugate (1:50, 4071, Cell Signaling Technology); β-catenin (1:200, 610153, BD); p-histone H3S10 (1:1,000, 06-570, Millipore); PTEN (1:50, 2551, home-made).

Secondary antibodies used were as follows: goat anti-mouse Alexa Fluor 488 (1:250, A11001, Invitrogen), goat anti-mouse Alexa Fluor 594 (1:250, A11005, Invitrogen), goat anti-mouse IgG1 Alexa Fluor 488 (1:250, A21121, Invitrogen), goat anti-rabbit Alexa Fluor 488 (1:250, A11008, Invitrogen), goat anti-rabbit Alexa Fluor 594 (1:250, A11012, Invitrogen).

*PTEN rabbit polyclonal antibody* 2551 for immunostaining of tissue was generated by immunizing a rabbit with GST-tagged recombinant C-domain mouse PTEN (amino acids 239-403) expressed in Escherichia Coli, purifying the IgG from antisera, and testing specificity by immunoblotting against whole cell lysates from MEFs positive and negative for PTEN (Supplementary Fig. 5c). Antibodies were further tested by IF on immortalized *Pten*Flox/Flox MEFs with stably transfected lentivirus TSIN-empty vector or TSIN-Cre (Supplementary Fig. 5d).

Cells and tissue were imaged with a laser-scanning microscope (LSM880, Carl zeiss) as described under "Mitotic Analyses" with either a 40× water-immersion lens or a 100× Alpha Plan-APO 100×/1.46 Oil DIC VIS lens, or with a fluorescent microscope (Olympus BX63, with standard 10×, 20× and 40× lenses).

## qRT-PCR

To test for mRNA stability, *Pten*+/+, *Pten*S380A/A, and *Pten*S380D/D MEFs were seeded in 6-well plates at $3 \times 10^5$ cells/well. 24 h after seeding, media was aspirated and Actinomycin D (Invitrogen, cat. #A7592; 5 μg/ml) containing media was added to the wells. RNA was collected at 0, 1, 2, 4, 6, and 8 h after addition by directly lysing the cell monolayer using standard procedures according to the Qiagen RNeasy Mini Kit (cat. #74104). cDNA synthesis was performed using random hexamers according to manufacturer's instructions (Invitrogen cat. #18080051) and qRT-PCR using SYBR mastermix was performed as described (Baker et al., NCB 2008). Primer sequences used: Pten – For 5′- TGGA TTCGACTTAGACTTGACCT −3′, Rev 5′- GCGGTGTCATAATGTCTCT CAG −3′; TBP – For 5′- GGCCTCTCAGAAGCATCACTA −3′, Rev 5′- GC CAAGCCCTGAGCATAA −3′.

## RNAseq library preparation, sequencing and bioinformatic analyses

RNA extraction from prostates (RNeasy Mini kit, #74104, QIAGEN) was performed according to the manufacturer's instructions. RNA quality and quantity were assessed using Agilent Bioanalyzer RNA 6000 Pico chips (5067-1513, Agilent Technologies). Equal amounts (300 ng) of high-quality RNA were subjected to library preparation using the TruSeq RNA Library Prep kit v2 (RS-122-2001, Illumina) according to the manufacturer's instructions. Libraries were sequenced following Illumina's standard protocol using the Illumina cBot and HiSeq 3000/4000 PE Cluster Kit. Flow cells were sequenced as 100 × 2 paired-end reads on an Illumina HiSeq 4000 using HiSeq 3000/4000 sequencing kit and HCS v.3.3.20 collection software. Base-calling was performed using Illumina's RTA v.2.5.2 software. RNA sequencing was performed at the Mayo Clinic Center for Individualized Medicine Medical Genomics Facility (Mayo Clinic, Rochester, Minnesota). Fastqfiles of paired-end RNA-Seq reads were aligned with Tophat v.2.0.14[67] against the UCSC reference genome mm10 (http://genome.ucsc.edu/cgi-bin/hgGateway?db=mm10) using Bowtie v.2 2.2.6 with default settings. Gene level counts from read pairs were obtained using FeatureCounts v.1.4.6 from the SubRead package[68] and gene models from the UCSC mm10 annotation. Differential expression analysis was performed using R package DESeq2 v.1.10.1 after removing genes with average raw counts <10[69]. Genes with lfcMLE (unshrunken maximum likelihood estimate of $\log_2$ fold change produced by DESeq2) >0.45 or <−0.45, and FDR < 0.05 were considered significantly differentially expressed. Gene Set Enrichment Analysis (GSEA)[70] was performed to detect enriched pathways against mouse gene sets from Enrichment Map

using gene lists ranked by lfcMLE, in descending order. Over-representation analysis for transcription regulatory targets of individual TFs was performed using the Fisher's exact test method for selected gene lists against the mouse gene sets from ENCODE (https://www.encodeproject.org/) and MSigDB (https://www.gsea-msigdb.org/gsea/msigdb/mouse/collections.jsp) collections. For over-representation analysis on gastric and prostate cancers in the TCGA database, we obtained transcriptomic data from the GDC portal by using R package TCGABiolinks as raw counts. Samples were normalized using DESeq2 and grouped based on normalized *PDZK1* expression: samples with *PDZK1* level lower than 15 percentiles were considered low-*PDZK1* and those higher than 85 percentiles high-*PDZK1*. Differential *PTEN* expression analysis between these two groups was performed using DESeq2. For assessing enrichment of LEF1-activated genes between the two groups, the Fisher's exact test was used to determine overrepresented gene sets in significantly upregulated genes ($p < 0.05$ and log fold change $\geq 1$) in the *PDZK1*-low group. Morpheus (Broad Institute) was used to generate heatmaps.

### Statistical analyses
GraphPad Prism software version 9.3.1 was used for statistical analyses. Error bars represent error of the mean. Statistical significance was determined using two-tailed Fisher's exact test, two-tailed unpaired *t*-test, one-way analysis of variance (ANOVA) with Dunnett's or Tukey's multiple comparisons test. *P* values are indicated in the figures and can also be found in the Source Data file. Sample sizes for all animal studies were chosen based on previously published studies in which differences were observed. The experiments were not randomized, and the investigators were not blinded.

### Reporting summary
Further information on research design is available in the Nature Portfolio Reporting Summary linked to this article.

## Data availability
The RNAseq data generated in this study have been deposited in the NCBI GEO database under accession code GSE206157. Open access transcriptomic data for our over-representation analyses on gastric and prostate cancers were obtained via Genomic Data Commons Data Portal (NCI) and downloaded as raw counts using R package TCGA-Biolinks. All remaining data are available within the paper, Supplementary Information, or Source data file. Source data are provided with this paper. Biological materials generated in this paper are available from the corresponding author. Source data are provided with this paper.

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

## Acknowledgements

We are grateful to Ines Sturmlechner and Cynthia Sieben for advice on bioinformatics methods used and the Mayo Clinic Gene Knockout and Transgenics Core for generating *Pten* mutant mouse strains. This work was supported by R01 CA242023 (J.M.v.D.).

## Author contributions

J.M.v.D. and J.H.v.R. conceived the project and J.H.v.R. coordinated the execution. M.H. developed and implemented gene editing methods for the generation of Pten hypomorphic alleles with various levels of PTEN reduction, and J.H.v.R., R.O.F.V. and I.C. maintained mouse cohorts. J.H.v.R. executed all tumor studies, tissue isolations, and mitosis-related studies. J.H.v.R. and I.C. generated MEF lines. K.B.J. conducted all western blot analyses, fractionation studies and immunoprecipitations and J.M.v.D. conducted in vitro binding studies. J.H.v.R and R.O.F.V. sectioned and stained whole prostates, R.O.F.V. performed PIN lesion scoring. K.B.J. generated PTEN antibody, which was validated by K.B.J. and J.H.v.R., and D.J.B. conducted qRT-PCR analyses. J.H.v.R. conducted all immunostainings. J.H.v.R and K.B.J. made RNA library. C.Z., J.H.v.R. and I.C. performed bioinformatic analyses. J.H.v.R. and J.M.v.D. wrote the paper. All authors edited the paper. H.L. supervised bioinformatic analyses and J.M.v.D. oversaw all other aspects of the study with input and support of D.J.B. J.M.v.D. obtained funding for the project.

## Competing interests

J.M.v.D. is a co-founder of and holds equity in Unity Biotechnology and Cavalry Biosciences. D.J.B. is a shareholder and co-inventor on patent applications licensed to or filed by Unity Biotechnology. This research has been reviewed by the Mayo Clinic Conflict of Interest Review Board and is being conducted in compliance with Mayo Clinic Conflict of Interest policies. All other authors declare no competing interests.
