## [Peer Review File · Nature Communications]

Hyperphosphorylated PTEN exerts oncogenic propertiesREVIEWER COMMENTS

Reviewer #1 (Remarks to the Author):

In this manuscript, by introducing subtle mutations in the PTEN C-tail domain, Janine H van Ree et al. show that C-tail mutant mice with an S380A substitution display low levels of PTEN and increased AKT signaling but no tumor predisposition, while mice carrying an S380D phosphomimetic mutation exhibit b-catenin hyperactivity in addition to PTEN instability and AKT hyperactivity. They show that aberrant b-catenin drives prostate neoplasia in these mice and provide data to suggest that this mechanism may be clinically relevant in gastric cancers. These data suggest that C-tail hyperphosphorylation creates oncogenic PTEN and is a potential target for anti-cancer therapy. This is an interesting manuscript that explored the oncogenic role of PTEN with C-tail hyperphosphorylation. Overall, this manuscript is suitable for publication in Nature Communications, however there are several major points below that need to be addressed in order to strengthen the significance of the findings supporting the authors' claims.

(1) It has been widely reported that C-tail phosphorylation increases PTEN stability. However, as shown in Fig. 5a, western blot analysis of prostate lysates from 2-month-old mice revealed that PTEN levels in PtenS380D/D mice were similarly reduced as in PtenS380A/A and Pten+/- mice. The authors should test the half-life of PTEN mRNA and the stability of PTEN protein in PtenS380D/D mice.

(2) Several N-terminal extended PTEN isoforms such as PTEN-Long (or PTEN α), and PTEN β has recently been identified, which are translated from an upstream CUG codon within PTEN 5'UTR. Considering that PTEN-Long (or PTEN α), and PTEN β contain the intact sequence of the PTEN protein, the authors should also test the effect of these PTEN isoforms with C-tail hyperphosphorylation on WNT/b-catenin signaling.

(3) The S380 residue is hyperphosphorylated in human gastric cancers, which is correlated with poor clinical prognosis. In the current study, they show that PtenS380D/D mice have neoplastic growth in prostate. Is there neoplastic growth in gastric tissue? Also, they should test the phosphorylation level of S380 in prostate cancer tissues.

Reviewer #2 (Remarks to the Author):

The authors present interesting and intriguing results, with what appears to be robust experimental and methodological design. My general evaluation is that the results presented are based on well-carried out experiments, a variety of mouse models generated and an attempt to translate the findings to the human pathology. However, I find the manuscript to lack a consistent flow of results, and that it contains different evidence that should be developed independently. To provide a more detailed evaluation of the manuscript, I would like to base my initial comments on the introductory discussion of the authors, and contextualize these statements with the results presented.

Statements taken from the discussion section:

First, we demonstrate that AKT hyperactivity caused by PTEN insufficiency is not necessarily oncogenic.

This is, in my view, an overstatement. The authors based their conclusions on a series of PTEN mutants that show reduced stability (yielding lower levels) but with unclear functional consequences on the retained PTEN moiety. It could well be that the oncogenic signaling elicited by de-inhibition of PI3K is being counteracted by an exacerbated function retained by the PTEN mutant forms (A/A, D4/D4, D/D).

Second, we show that substantial reductions in PTEN level do not necessarily increase tumor susceptibility.

As indicated above, this is based on the analysis of the mutants, and the remaining PTEN moiety is not WT. Therefore, the counterbalancing activities of the remaining mutant protein could alter the conclusion of the authors.

Third, we provide evidence that initiation of neoplastic growth in prostates of Pten haploinsufficient

mice involves loss-of-function of both alleles.

The data provided to support this statement are insufficient, but if further developed it could represent an interesting information for the field. This constitutes a project on its own, and in the context of this study it introduces confusion.

Fourth, we uncovered that aberrant PTEN C-tail phosphorylation in mice drives prostate neoplasia through oncogenic activation of WNT signaling and provide evidence to suggest that this mechanism is relevant to human gastric cancers.

This statement is not supported by data. The authors show that any of the mutations generated (A/A, D/D, D4/D4) destabilize PTEN. Only two of these mutations are analyzed in a whole-organism level (phosphodead and D4) and are shown convincingly not to elicit a tumorigenic phenotype despite lower PTEN level and increased AKT activity. However, the S380D mutant is not studied in these conditions, and the effects presented in prostate pathology are marginal. In addition, the S380D mutant also destabilizes PTEN and activates AKT. The most plausible explanation for these results is that perturbation of S380 or that region exerts detrimental consequences on PTEN regardless of the phosphorylation of S380. The data on WNT signaling is intriguing, but the cross with CTNNB1 hets showing that reduces nuclear beta-cat and the already mild phenotype (2% of tubules with grade 2 PIN in the best scenario), is not meaningful to support such a strong conclusion.

Additional comments:

- I restate that the authors should consider a stronger focus on gastric tissue, since they argue that this is the context where the phosphorylation is relevant according to prior studies.
- The data of mitotic defects in figure 1 is ignored throughout the rest of the study, and its relevance is unclear for this study. For the following statement made by the authors: S380 phosphorylation is a requirement for DLG1-mediated centrosomal accumulation of PTEN; the S380D mutant should also be analyzed for these parameters.
- The analysis of the hypomorphic, the different genetic backgrounds and the LOH phenotype do not fit in the flow of the study and make the reading confusing. In addition, further analyses would be required (proper LOH analysis, for example) to support the conclusions.
- The PIN incidence and grade in the S380D does not warrant a strong statement on its tumorigenic properties. One could argue that S to D mutation inactivates PTEN, which added to its destabilization and AKT activation increases tumorigenic capacity. In this context, the mutation would exert a passive function, leading to loss of activity, rather than an active function, "driving" PIN formation.
- The IF in Fig 5e is not very clear.
- The evidence of WNT activation in the S380D mutant is not very robust. In addition, the correlations of PDZK1 with Beta cat activity lead to overstatements, as changes in PDZK1 expression will lead to alterations in other proteins that regulate Beta cat. A proper analysis in this direction would be ascertaining PTEN S380 phosphorylation in tumor specimens with available RNA Seq data, so that beta cat activity can be analyzed (or alternatively, nuclear Beta Cat staining quantification in the same tissues).
- Combining S380D mutant mice with Ctnnb1 het mice to show that nuclear beta cat staining decreases is not meaningful, and it is expected from the reduced beta cat abundance and the lower immunoreactivity. In addition, since the PIN phenotype in these mice is very mild, the reduction in the Ctnnb1 het is not very meaningful.

Reviewer #3 (Remarks to the Author):

van Ree et al presented a study that seeks to understand how the C-terminal tail of PTEN and phosphorylation of Ser380 contribute to tumorigenesis and WNT signaling in mice. Previous work assigned key regulatory functions to phosphorylation of PTEN's C-terminal tail at Ser380, Thr382, Thr383, and Ser385, which inhibits its activity, reduces its plasmid membrane binding, and increases its stability. In this role, phosphorylation of the C-terminal tail guides PTEN's decision between activity and stability. Therefore, the authors took a rational approach to understanding how dysregulation of PTEN C-terminal phosphorylation can influence signaling and contribute to cancer.

The major findings of their work indicate that activation of AKT signaling doesn't necessarily lead to tumorigenesis, reduction in PTEN levels doesn't necessarily lead to tumors, and hyperphosphorylation of PTEN Ser380 leads to neoplastic growth in the prostate by an increase in WNT signaling, perhaps through its nuclear interaction with B-catenin. With this said, the paper was interesting, but we have several concerns that we believe should be addressed, as stated below.

1. As stated above, phosphorylation of a cluster of residues on the C-terminal tail regulates PTEN's activity and stability; however, the authors only focused on Ser380. Each site on the C-terminal tail (Ser380, Thr382, Thr383, and Ser385) has a similar contribution to the closed state, so not knowing their phosphorylation status could complicate the meaning of their mutant results (AKT activation vs. PTEN activity/stability and tumorigenesis). It would be helpful to know if other sites on the tail are phosphorylated and how this may influence their observed outcomes.

2. Data from the prostate showed reduced levels of PTEN S380D (Fig. 5b). One would expect that phosphorylation of these residues would increase stability, so it is surprising to see a reduction in the PTEN protein level. Is this because the mutation reduces total tail phosphorylation, hence decreasing PTEN's stability, or is the Asp mutation not a good mimic for Ser380 phosphorylation? With this said, AKT activity is increased with this mutation, consistent with S380D being autoinhibitory. Knowing the phosphorylation status of the other residues would help understand this unexpected result.

3. The authors have proposed that PTEN directly engages with B-catenin in the nucleus when Ser380 is hyperphosphorylated as a possible mechanism for neoplastic growth in the prostate by immunoprecipitation. Immunoprecipitations cannot report on direct interactions but may indicate that two proteins are found in a larger complex. Furthermore, the band intensity was very weak as compared to the input, and the IgG control IP was very intense, which is concerning. Therefore, to prove a direct interaction between PTEN and B-catenin, the authors should use purified proteins to validate this interaction.

POINT-BY-POINT RESPONSES TO THE COMMENTS OF THE THREE REVIEWERS

Our responses are in blue text.

Reviewer #1 (Remarks to the Author):

In this manuscript, by introducing subtle mutations in the PTEN C-tail domain Janine H van Ree et al. show that C-tail mutant mice with an S380A substitution display low levels of PTEN and increased AKT signaling but no tumor predisposition, while mice carrying an S380D phosphomimetic mutation exhibit b-catenin hyperactivity in addition to PTEN instability and AKT hyperactivity. They show that aberrant b-catenin drives prostate neoplasia in these mice and provide data to suggest that this mechanism may be clinically relevant in gastric cancers. These data suggest that C-tail hyperphosphorylation creates oncogenic PTEN and is a potential target for anti-cancer therapy. This is an interesting manuscript that explored the oncogenic role of PTEN with C-tail hyperphosphorylation. Overall, this manuscript is suitable for publication in Nature Communications, however there are several major points below that need to be addressed to strengthen the significance of the findings supporting the authors' claims.

We appreciate that the reviewer is supportive of our study. The proposed experiments were very insightful in terms of understanding (i) the impact of our C-tail mutations on the stability of PTEN and its isoforms, and (ii) the potential relevance of our mouse study mice to human prostate cancer.

(1) It has been widely reported that C-tail phosphorylation increases PTEN stability. However, as shown in Fig. 5a, western blot analysis of prostate lysates from 2-month-old mice revealed that PTEN levels in *Pten*^{S380D/D} mice were similarly reduced as in *Pten*^{S380A/A} and *Pten*^{+/-} mice. The authors should test the half-life of PTEN mRNA and the stability of PTEN protein in *Pten*^{S380D/D} mice.

Response: We thank the reviewer for recommending these insightful experiments. To determine the extent to which the low PTEN^{S380D} levels are due to reduced protein stability, we performed cycloheximide chase experiments on *Pten*^{+/+} and *Pten*^{S380D/D} MEFs. These experiments revealed that PTEN^{S380D} is less stable than PTEN, just like PTEN^{S380A}. We have incorporated these results in revised Fig. 5c and Supplementary Fig. 6d and in the main text on page 7 (lines 1-2).

The requested actinomycin D chase experiments revealed that *Pten*^{S380D/D} MEFs contain normal amounts of *Pten* transcripts and that these transcripts are as highly stable as those derived from the wildtype *Pten* locus. The same was also true for *Pten* transcripts of *Pten*^{S380A/A} MEFs. Please see Supplementary Fig. 6e and the main text on page 7 (lines 1-3).

(2) Several N-terminal extended PTEN isoforms such as PTEN-Long (or PTEN α) and PTEN β has recently been identified, which are translated from an upstream CUG codon within PTEN 5'UTR. Considering that PTEN-Long (or PTEN α) and PTEN β contain the intact sequence of the PTEN protein, the authors should also test the effect of these PTEN isoforms with C-tail hyperphosphorylation on WNT/b-catenin signaling.

Response: This is another terrific point. Indeed, multiple studies have now reported on the existence of two low-abundance PTEN isoforms with 146 and 173 amino-acid terminal extensions. These isoforms arise due to the presence of alternative translation start sites in *PTEN* mRNA¹⁻⁵. PTEN α and PTEN β isoforms typically appear on western blots probed with PTEN Rabbit mAb 138G6 from Cell Signaling Technologies as minor bands of ~76 and ~72 kDa, respectively (requiring extensive separation and long exposure times). In using cell fractionation in combination with western blotting, Liang and colleagues demonstrated that PTEN α and PTEN β are enriched in the membrane and nuclear fractions, respectively^{1,2}.

We used the latter insights to examine the effect of the S380D and S380A mutations on the two long PTEN isoforms. Specifically, we prepared cytoplasmic, membrane, and nuclear fractions of *Pten*^{+/+}, *Pten*^{S380D/D} and *Pten*^{S380A/A} MEFs and analyzed these by western blotting using PTEN Rabbit mAb 138G6 (n=3 MEF lines per genotype). As previously reported, we found PTEN α to be enriched in the membrane fraction of *Pten*^{+/+} MEFs. Similar amounts of PTEN α were also membrane-associated in *Pten*^{S380D/D} and *Pten*^{S380A/A} MEFs, indicating that the S380D and S380A mutations do not negatively impact the stability of this isoform. The same seemed true for the PTEN β in the nuclear fractions of *Pten*^{S380D/D} and *Pten*^{S380A/A} MEFs, taking in consideration that the S380D and S380A mutations respectively decrease and increase PTEN nuclear localization. These findings are consistent with the observation that *Pten* transcript levels are not affected by the presence of the mutations that created the S380A and D substitutions. We have incorporated these new data in Fig. 5d and Supplementary Fig. 6f and in the main text on page 7 (lines 3-12).

(3) The S380 residue is hyperphosphorylated in human gastric cancers, which is correlated with poor clinical prognosis. In the current study, they show that *Pten*^{S380D/D} mice have neoplastic growth in prostate. Is there neoplastic growth in gastric tissue? Also, they should test the phosphorylation level of S380 in prostate cancer tissues.

Response: Our initial efforts of exploring the potential relevance of our data in mice, suggesting that PTEN C-tail hyperphosphorylation drives neoplastic growth in the prostate by promoting the nuclear accumulation of β -catenin, focused on human gastric tumors characterized by hyperphosphorylation of PTEN S380 using RNA-sequencing data available from TCGA⁶⁻⁸. In this analysis we used low expression of the PTEN C-tail phosphorylation inhibitor PDZK1 as a marker for S380 hyperphosphorylation⁷. We stratified TCGA gastric tumors based on *PDZK1* mRNA levels into *PDZK1* low (bottom 15%) and *PDZK1* high (top 15%) cohorts, compared these two groups for differentially expressed genes, and then performed an overrepresentation analysis for target genes of LEF1, a β -catenin-dependent transcription factor associated with gastric cancer progression⁹. We found that low *PDZK1* indeed correlated with LEF1 hyperactivity, whereas high *PDZK1* did not, supporting the idea that PTEN C-tail hyperphosphorylation drives pro-tumorigenic WNT signaling in gastric cancers in which *PDZK1* expression is compromised. To explore whether the same might hold true for human prostate cancers, we performed the same analysis on TCGA prostate cancers. We found that low *PDZK1* expression here also correlated with LEF1 hyperactivity, a known driver of prostate tumorigenesis as previously demonstrated by Li et al.¹⁰. We have included these new data in Fig. 8b and in the main text on page 9 (lines 17-21). Please see also our validation that prostate tumors with low *PDZK1* contain an abundance of *PTEN* transcripts in Supplementary Fig. 9b.

To complement the above analysis, we had also requested human prostate cancer sections for P-S380 immunohistochemistry but unfortunately, due to COVID-19-related staffing shortages and major backlogs in the Mayo Clinic core facility tasked with providing such samples, even

after 3 months they are still not able to provide us with a timeline for when we could expect the requested samples. Referring to the question as to whether *Pten*^{S380D/D} mice are prone to gastric tumors, there were limitations with regards to mice. The *Pten*^{S380D/D} strain was specifically generated to look at PIN lesions as we had indicated in the text, and we did not set up a cohort for long-term observation. We agree that it will be interesting as a future extension of the current study to set up such a cohort to evaluate the extent to which the *Pten*^{S380D} mutation predisposes other tissues to neoplastic growth.

Reviewer #2 (Remarks to the Author):

The authors present interesting and intriguing results, with what appears to be robust experimental and methodological design. My general evaluation is that the results presented are based on well-carried out experiments, a variety of mouse models generated and an attempt to translate the findings to the human pathology.

However, I find the manuscript to lack a consistent flow of results, and that it contains different evidence that should be developed independently. To provide an more detailed evaluation of the manuscript, I would like to base my initial comments on the introductory discussion of the authors, and contextualize these statements with the results presented.

We were happy that the reviewer felt the study was interesting and well-designed, and we appreciated the thoughtful and incisive comments regarding the interpretation and presentation of our data, which we have worked to address in the revised manuscript by further experimentation or changes to the text and figures.

Statements taken from the discussion section:

First, we demonstrate that AKT hyperactivity caused by PTEN insufficiency is not necessarily oncogenic. This is, in my view, an overstatement. The authors based their conclusions on a series of PTEN mutants that show reduced stability (yielding lower levels) but with unclear functional consequences on the retained PTEN moiety. It could well be that the oncogenic signaling elicited by de-inhibition of PI3K is being counteracted by an exacerbated functions retained by the PTEN mutant forms (A/A, D4/D4, D/D).

Second, we show that substantial reductions in PTEN level do not necessarily increase tumor susceptibility. As indicated above, this is based on the analysis of the mutants, and the remaining PTEN moiety is not WT. Therefore, the counterbalancing activities of the remaining mutant protein could alter the conclusion of the authors.

Response: Although controlling AKT signaling is considered PTEN's major tumor suppressive function in the literature, several other tumor suppressive mechanisms have also been described. Indeed, it is difficult to exclude that these other mechanisms are reinforced to counteract AKT hyperactivity. We have contracted the first two statements into a single statement that considers the fact that there could be counterbalancing activities at work. It reads: "First, we demonstrate that *Pten* mutations that substantially reduce protein level and cause AKT hyperactivity are not necessarily tumorigenic." Please see also changes to the main text on pages 9 (lines 31-33). Please see also page 10 (lines 14-24).

Third, we provide evidence that initiation of neoplastic growth in prostates of *Pten* haplo-insufficient mice involves loss-of-function of both alleles. The data provided to support this statement

are insufficient, but if further developed it could represent an interesting information for the field. This constitutes a project on its own, and in the context of this study it introduces confusion.

Response: We agree that to claim PTEN loss of heterozygosity (LOH) one would provide (epi)genetic evidence, which is difficult in the case of prostate lesions. However, using high AKT S473 phosphorylation as a marker for PTEN loss in immunostaining experiments, we show that from the earliest stages of neoplastic growth this marker is present in combination with reduced PTEN staining. We believe that it is reasonable to conclude based on these findings that the residual wildtype allele in *Pten*^{+/-} mice is functionally compromised at the onset of neoplastic growth. To address the reviewer's concern that our statement is confusing we have changed it. It now reads: "Second, we provide evidence to suggest that initiation of neoplastic growth in prostates of *Pten* haploinsufficient mice requires additional loss of PTEN function".

Please see the main text on page 9 (lines 33-35). See also the more extensive discussion on this topic on page 11 (lines 8-24). Please see also relevant changes to the text in the results section on pages 5 (lines 7-15 and 42-44) and 6 (lines 1-7).

Fourth, we uncovered that aberrant PTEN C-tail phosphorylation in mice drives prostate neoplasia through oncogenic activation of WNT signaling and provide evidence to suggest that this mechanism is relevant to human gastric cancers.

This statement is not supported by data. The authors show that any of the mutations generated (A/A, D/D, D4/D4) destabilize PTEN. Only two of these mutations are analyzed in a whole-organism level (phosphodead and D4) and are shown convincingly not to elicit a tumorigenic phenotype despite lower PTEN level and increased AKT activity.

However, the S380D mutant is not studied in these conditions, and the effects presented in prostate pathology are marginal. In addition, the S380D mutant also destabilizes PTEN and activates AKT. The most plausible explanation for these results is that perturbation of S380 or that region exerts detrimental consequences on PTEN regardless of the phosphorylation of S380.

Response: The reviewer points out that the statement that "we uncovered that aberrant PTEN C-tail phosphorylation in mice drives prostate neoplasia through oncogenic activation of WNT signaling and provide evidence to suggest that this mechanism is relevant to human gastric cancers" is incorrectly stated. We agree and apologize for the imprecision. We intended this statement to be based on our findings in *Pten*^{S380D/D} mice, but we did not realize that it could also be taken as to pertain to *Pten*^{Δ4/Δ4} and *Pten*^{S380A/A} mice, given that these mice also have aberrant PTEN C-tail phosphorylation. We have now corrected the statement to exclude these two mutants and it now reads: "Third, we provide data to suggest that hyperphosphorylation of PTEN C-tail residue S380 drives prostate neoplasia in mice through oncogenic activation of WNT signaling and that this mechanism is potentially relevant to human cancers. Please see the changes to the text on page 9 (lines 35-37).

The reviewer also comments that we did not comprehensively explore the tumor phenotype of *Pten*^{S380D/D} mice. Indeed, we specifically generated this strain to determine its predisposition to neoplastic growth in the prostate, the organ our study had come to focus on at that time. Only cohorts of mice for this specific purpose were generated. We agree with the reviewer that given what we have learned from studying the prostate of *Pten*^{S380D/D} mice, it would be interesting to generate larger cohorts of mice to conduct a longer-term follow-up study in the future that focuses on other organs.

Whereas the reviewer finds that the prostate pathology observed in *Pten*^{S380D/D} mice is marginal, we believe our observed phenotype is biologically relevant. While it is evident that the tumor phenotype of *Pten*^{+/-} mice is more profound, 6-month-old *Pten*^{S380D/D} mice have 10-fold more PIN II than age-matched *Pten*^{+/+} mice (2% vs 0.2% of tubules). Prostates of 7 out of 10 *Pten*^{S380D/D} mice analyzed (Fig. 5f and Fig. 7c) contained PIN III lesions, a more advanced lesion type not observed in any of the corresponding *Pten*^{+/+} prostates. Moreover, we explored the underlying mechanism of PIN lesion formation in *Pten*^{S380D/D} mice by testing the hypothesis that PTEN^{S380D} might exert its tumor-promoting effect through β -catenin hyperactivity. This hypothesis was based on RNA-seq data from 2-month-old mice, which at that age do not show evidence of PIN lesion formation. We comprehensively tested the hypothesis in the following ways: (1) by using 2 independent approaches to measure β -catenin activity in prostates (IF-based assessment of nuclear β -catenin accumulation and a reporter transgene for β -catenin activity, Fig. 6e,f); (2) by inactivating a single *Ctnnb1* allele in *Pten*^{S380D/D} mice and testing whether this inhibits PIN lesion formation (Fig. 7c); (3) by examining PTEN- β -catenin protein-protein interaction in the prostate and how S380 mutation alters this (Fig. 7b). In each case, we obtained data that support our hypothesis. Furthermore, in response to the reviewer's comment, we have included an extended analysis of PIN lesions formation on 9-month-old *Pten*^{S380D/D} and *Pten*^{+/+} males (Supplementary Fig. 6g). This analysis shows a 10-fold increase in PIN III lesions in *Pten*^{S380D/D} mice between 6 and 9 months.

Additional comments:

- I restate that the authors should consider a stronger focus on gastric tissue, since they argue that this is the context where the phosphorylation is relevant according to prior studies.

Response: This comment aligns with reviewer 1's request to examine whether *Pten*^{S380D/D} mice are predisposed to stomach cancer and whether PTEN hyperphosphorylation and the consequent hyperactive WNT signaling observed in mice might be relevant to both human stomach and prostate cancer. Please see point 3 of reviewer 1 on how we have addressed this point.

- The data of mitotic defects in figure 1 is ignored throughout the rest of the study, and its relevance is unclear for this study. For the following statement made by the authors: S380 phosphorylation is a requirement for DLG1-mediated centrosomal accumulation of PTEN; the S380D mutant should also be analyzed for these parameters.

Response: Indeed, we did not address whether the mitotic defects observed in *Pten*^{S380A/A} and *Pten* ^{$\Delta 4/\Delta 4$} mice extend to *Pten*^{S380D/D} mice. We agree with the reviewer that this is important to include this assessment and we have done so. We found that the key mitotic phenotypes of *Pten*^{S380A/A} and *Pten* ^{$\Delta 4/\Delta 4$} MEFs were also observed in *Pten*^{S380D/D} MEFs, including aneuploidization, slow spindle pole movement and formation of non-perpendicular mitotic spindles. Please see Supplementary Fig. 8 and the main text on page 7 (lines 16-18).

With regards to the reviewer's comment that the mitotic data seem to lack relevance and perhaps do not benefit the flow of results we have greatly shortened the main results section regarding this topic and included the more detailed analysis as a Supplementary Note. We believe the mitotic data have relevance in the context of the reviewer's point about the potential reinforcement of other tumor suppressive functions that might offset the increase in AKT activity in C-tail mutant mice. Faithful chromosome segregation has been identified as one of PTEN's additional tumor suppressive functions beyond controlling AKT activity. Our data shows that this particular tumor suppressive function of PTEN is impaired rather than reinforced. While this does not

exclude the possibility that other perhaps more prominent tumor suppressive functions are reinforced, it at least provides some initial insight. Please see our discussion on this topic in the main text on page 10 (lines 14-24).

- The analysis of the hypomorphic, the different genetic backgrounds and the LOH phenotype do not fit in the flow of the study and make the reading confusing. In addition, further analyses would be required (proper LOH analysis, for example) to support the conclusions.

Response: We did not articulate the rationale for the abovementioned experiments well enough and we apologize for the confusion this created. We have now made various corrective changes to the text. Regarding the rationale for the hypomorphic studies, please see main text on page 5 (lines 7-15). As mentioned above, we agree that to claim PTEN LOH one would need to provide (epi)genetic evidence, which is difficult in the case of prostate lesions. However, using high AKT S473 phosphorylation as a marker for PTEN loss in immunostaining experiments, we show that from the earliest stages of neoplastic growth this marker is present in combination with reduced PTEN staining. We believe that it is reasonable to conclude based on these findings that the residual wildtype allele in *Pten*^{+/-} mice is functionally compromised at the onset of neoplastic growth. Please see the main text on page 9 (lines 33-35). See also the more extensive discussion on this topic on page 11 (lines 8-24). Please see also relevant changes to the text in the results section on pages 5 (lines 7-15) and 6 (lines 1-7).

- The PIN incidence and grade in the S380D does not warrant a strong statement on its tumorigenic properties. One could argue that S to D mutation inactivates PTEN, which added to its destabilization and AKT activation increases tumorigenic capacity. In this context, the mutation would exert a passive function, leading to loss of activity, rather than an active function, “driving” PIN formation.

Response: We find that the *Pten*^{S380A/A} and *Pten*^{Δ4/Δ4} mice are not tumor prone despite PTEN destabilization and increased AKT signaling, which is why it is unlikely that these same defects would be drivers of PIN lesions in *Pten*^{S380D/D} mice. Importantly, if loss of PTEN function was to drive neoplastic growth in *Pten*^{S380D/D} mice, it would be difficult to explain how that inactivation of a single *Cttnb1* allele in *Pten*^{S380D/D} mice would normalize PIN lesion formation to rates observed in wildtype mice.

- The IF in Fig 5e is not very clear.

Response: Agreed, we have replaced the original image of the *Pten*^{-/A} lesion with a clearer one. Please see revised Fig. 5h. The correlation between decreased PTEN and increased P-AKT S473 is now clear for both genotypes.

- The evidence of WNT activation in the S380D mutant is not very robust. In addition, the correlations of PDZK1 with Beta cat activity lead to overstatements, as changes in PDZK1 expression will lead to alterations in other proteins that regulate Beta cat. A proper analysis in this direction would be ascertaining PTEN S380 phosphorylation in tumor specimens with available RNA Seq data, so that beta cat activity can be analyzed (or alternatively, nuclear Beta Cat staining quantification in the same tissues).

Response: The evidence of WNT activation in the S380D mutant is based on data from a diverse set of experiments. First, the initial indication that WNT signaling might be hyperactive in *Pten*^{S380D/D} prostates came from unbiased analysis of transcriptomic data of prostates of 2-month-old mice of *Pten*^{S380A/A}, *Pten*^{S380D/D}, and *Pten*^{+/+} mice. Second, we used two distinct

approaches to measure β -catenin hyperactivity in prostates, both indicating that β -catenin is hyperactive in *Pten*^{S380D/D} prostates. Third, inactivation of a single *Ctnnb1* allele in *Pten*^{S380D/D} mice normalizes PIN lesion formation to rates observed in wildtype mice. Fourth, we show that PTEN and β -catenin form a complex and that in the case of PTEN^{S380D} there is an enrichment of complex in the nuclear compartment.

Regarding the other comments, we agree with the reviewer that a reduction in PDZK1 expression can affect the phosphorylation status of proteins beyond PTEN. We have included a statement to this effect. Please see page 12 (lines 6-11). We also agree that studies into the relevance of PTEN C-tail phosphorylation to human cancer ideally would include studies correlating PTEN C-tail phosphorylation with β -catenin hyperactivity. As mentioned in our response to point 3 of reviewer 1, we tried to obtain human prostate cancer sections for P-S380 and β -catenin immunohistochemistry but unfortunately, due to COVID-19-related staffing shortages and major backlogs in the Mayo Clinic core facility tasked with providing such samples, after 3 months they were still not able to provide us with a timeline for when we could expect the requested samples. However, we were able to further explore human relevance by extending our bioinformatics analysis on gastric tumors to TCGA prostate tumors. We found that low *PDZK1* expression here also correlated with LEF1 hyperactivity (a known driver of prostate tumorigenesis as previously demonstrated by Li et al¹⁰). We have included these data in Fig. 8b and in the main text on page 9 (lines 17-21). Please see also our validation that prostate tumors with low *PDZK1* contain an abundance of *PTEN* transcripts in Supplementary Fig. 9b.

- Combining S380D mutant mice with *Ctnnb1* het mice to show that nuclear beta cat staining decreases is not meaningful, and it is expected from the reduced beta cat abundance and the lower immunoreactivity. In addition, since the PIN phenotype in these mice is very mild, the reduction in the *Ctnnb1* het is not very meaningful.

Response: The rationale for including the assessment of β -catenin nuclear localization is based on the finding that less nuclear accumulation of β -catenin is consistent with a reduction in PIN lesion formation in *Pten*^{S380D/D};*Ctnnb1*^{+/-}. As discussed, although the PIN phenotype in *Pten*^{S380D/D} is milder than that of *Pten*^{+/-} mice, it is highly reproducible and progressive and can therefore be considered meaningful.

Reviewer #3 (Remarks to the Author):

van Ree et al presented a study that seeks to understand how the C-terminal tail of PTEN and phosphorylation of Ser380 contribute to tumorigenesis and WNT signaling in mice. Previous work assigned key regulatory functions to phosphorylation of PTEN's C-terminal tail at Ser380, Thr382, Thr383, and Ser385, which inhibits its activity, reduces its plasmid membrane binding, and increases its stability. In this role, phosphorylation of the C-terminal tail guides PTEN's decision between activity and stability. Therefore, the authors took a rational approach to understanding how dysregulation of PTEN C-terminal phosphorylation can influence signaling and contribute to cancer.

The major findings of their work indicate that activation of AKT signaling doesn't necessarily lead to tumorigenesis, reduction in PTEN levels doesn't necessarily lead to tumors, and hyperphosphorylation of PTEN Ser380 leads to neoplastic growth in the prostate by an increase in WNT signaling, perhaps through its nuclear interaction with B-catenin. With this said, the paper was

interesting, but we have several concerns that we believe should be addressed, as stated below.

We were pleased to hear that the reviewer felt our study was interesting. The follow up experiments requested by the reviewer were instrumental in understanding the impact of the S380 mutations on the phosphorylation status of other C-tail residues and the nature of the PTEN- β -catenin interaction.

1. As stated above, phosphorylation of a cluster of residues on the C-terminal tail regulates PTEN's activity and stability; however, the authors only focused on Ser380. Each site on the C-terminal tail (Ser380, Thr382, Thr383, and Ser385) has a similar contribution to the closed state, so not knowing their phosphorylation status could complicate the meaning of their mutant results (AKT activation vs. PTEN activity/stability and tumorigenesis). It would be helpful to know if other sites on the tail are phosphorylated and how this may influence their observed outcomes.

Response: We thank the reviewer for raising this important point regarding the impact of the S380A and D mutations on the phosphorylation status of the other C-tail residues in the S380-T382-T383-S385 cluster. Indeed, based on cell culture experiments it has been suggested that the order in which residues in this cluster get phosphorylated is S385 > S380 > T383 > T382¹¹. Phosphorylation of these residues reportedly increases PTEN stability but reduces its phosphatase activity and plasma membrane localization. The prevailing view based on these data is that, when phosphorylated, the C-tail interacts with the C2 and phosphatase domains to form a "closed" conformation. This conformer is less susceptible to ubiquitin-mediated degradation and thus more stable, but less capable of interacting with proteins that target PTEN to the plasma membrane, causing it to be less catalytically active¹².

To explore the impact of the S380A and D mutations on the phosphorylation status of T382, T383 and S385 we performed western blot analysis on lysates of *Pten*^{S380A/A}, *Pten*^{S380D/D} and *Pten*^{+/+} prostates using phospho-specific antibodies against the S380-T382-T383 cluster and S385 alone. We also included the phospho-specific S380 antibody that we had used to confirm the lack of S380 phosphorylation in *Pten*^{S380A/A} MEFs. The latter antibody not only failed to detect PTEN^{S380A} but also PTEN^{S380D}. On the other hand, the phospho-specific antibody against S380-T382-T383 detected both PTEN^{S380A} and PTEN^{S380D}. The same was true for S385. For both antibodies, the reductions in phospho-specific signals that we observed when comparing PTEN^{S380A} (*Pten*^{S380A/A}) and PTEN^{S380D} (*Pten*^{S380D/D}) to PTEN (*Pten*^{+/+}) corresponded well with the reductions observed with regular (non-phospho-specific) PTEN antibodies. These data suggest that the S380A and S380D mutations had no major impact on the phosphorylation status of T382, T383, and S385, and that the observed changes in the properties of PTEN^{S380A} and PTEN^{S380D} seem to be a direct consequence of the altered S380 phosphorylation status. We have incorporated this experiment in revised Fig. 5a and Supplementary Fig. 6a-c, and in the main text on page 6 (lines 36-42).

2. Data from the prostate showed reduced levels of PTEN S380D (Fig. 5b). One would expect that phosphorylation of these residues would increase stability, so it is surprising to see a reduction in the PTEN protein level. Is this because the mutation reduces total tail phosphorylation, hence decreasing PTEN's stability, or is the Asp mutation not a good mimic for Ser380 phosphorylation? With this said, AKT activity is increased with this mutation, consistent with S380D being autoinhibitory. Knowing the phosphorylation status of the other residues would help understand this unexpected result.

Response: This point relates to point 1 (please see above for details). We find that the S380D mutation does not seem to reduce total C-tail phosphorylation. We agree that the result is

surprising considering earlier reports. One important difference with earlier reports is that ectopically expressed PTEN C-tail mutants were studied in cells expressing endogenous PTEN, whereas in our case cells contained no wildtype PTEN. Given that PTEN homodimerizes, this could be an important difference with regard to protein stability.

We further note that in response to comment 2 of reviewer 1 we looked at the impact of the S380 D and A mutations on the PTEN long isoforms generated through alternative translation initiation. Interestingly, our western blot data suggest that neither mutation alters the stability of the longer PTEN α and PTEN β isoforms. For further details, please see how we addressed comment 2 of reviewer 1.

3. The authors have proposed that PTEN directly engages with B-catenin in the nucleus when Ser380 is hyperphosphorylated as a possible mechanism for neoplastic growth in the prostate by immunoprecipitation. Immunoprecipitations cannot report on direct interactions but may indicate that two proteins are found in a larger complex. Furthermore, the band intensity was very weak as compared to the input, and the IgG control IP was very intense, which is concerning. Therefore, to prove a direct interaction between PTEN and B-catenin, the authors should use purified proteins to validate this interaction.

Response: To examine whether the interaction between PTEN and β -catenin is direct, we immunoprecipitated recombinant human PTEN protein (R&D systems) with an antibody against C-terminal PTEN and screened for coprecipitation of recombinant human β -catenin (Sino Biological) by western blot analysis. Indeed, β -catenin coprecipitated with PTEN whereas a corresponding control antibody (to detect MYC-tags) did not. We incorporated this experiment in revised Fig. 7a and in the text on page manuscript on page 8 (lines 29-30).

REFERENCES

- 1 Liang, H. *et al.* PTENbeta is an alternatively translated isoform of PTEN that regulates rDNA transcription. *Nat Commun* **8**, 14771, doi:10.1038/ncomms14771 (2017).
- 2 Liang, H. *et al.* PTENalpha, a PTEN isoform translated through alternative initiation, regulates mitochondrial function and energy metabolism. *Cell Metab* **19**, 836-848, doi:10.1016/j.cmet.2014.03.023 (2014).
- 3 Shen, S. M. *et al.* PTENalpha and PTENbeta promote carcinogenesis through WDR5 and H3K4 trimethylation. *Nat Cell Biol* **21**, 1436-1448, doi:10.1038/s41556-019-0409-z (2019).
- 4 Taylor, J. & Abdel-Wahab, O. PTEN isoforms with dual and opposing function. *Nat Cell Biol* **21**, 1306-1308, doi:10.1038/s41556-019-0405-3 (2019).
- 5 Zhang, C. *et al.* Furin extracellularly cleaves secreted PTENalpha/beta to generate C-terminal fragment with a tumor-suppressive role. *Cell Death Dis* **13**, 532, doi:10.1038/s41419-022-04988-2 (2022).
- 6 Yang, Z. *et al.* Reduced expression of PTEN and increased PTEN phosphorylation at residue Ser380 in gastric cancer tissues: a novel mechanism of PTEN inactivation. *Clin Res Hepatol Gastroenterol* **37**, 72-79, doi:10.1016/j.clinre.2012.03.002 (2013).
- 7 Zhao, C. *et al.* Loss of PDZK1 expression activates PI3K/AKT signaling via PTEN phosphorylation in gastric cancer. *Cancer Lett* **453**, 107-121, doi:10.1016/j.canlet.2019.03.043 (2019).
- 8 Yang, Z. *et al.* Phosphorylation and inactivation of PTEN at residues Ser380/Thr382/383 induced by *Helicobacter pylori* promotes gastric epithelial cell survival through PI3K/Akt pathway. *Oncotarget* **6**, 31916-31926, doi:10.18632/oncotarget.5577 (2015).
- 9 Zhou, L. Q., Li, S. H., Wu, Y. & Xin, L. Establishment of a prognostic model of ten transcription factors in gastric cancer. *Genomics* **113**, 4075-4087, doi:10.1016/j.ygeno.2021.10.009 (2021).

- 10 Li, Y. *et al.* LEF1 in androgen-independent prostate cancer: regulation of androgen receptor expression, prostate cancer growth, and invasion. *Cancer Res* **69**, 3332-3338, doi:10.1158/0008-5472.CAN-08-3380 (2009).
- 11 Fragoso, R. & Barata, J. T. Kinases, tails and more: regulation of PTEN function by phosphorylation. *Methods* **77-78**, 75-81, doi:10.1016/j.ymeth.2014.10.015 (2015).
- 12 Song, M. S. *et al.* Nuclear PTEN regulates the APC-CDH1 tumor-suppressive complex in a phosphatase-independent manner. *Cell* **144**, 187-199, doi:10.1016/j.cell.2010.12.020 (2011).

REVIEWERS' COMMENTS

Reviewer #1 (Remarks to the Author):

All the questions I concerned have been resolved in the POINT-BY-POINT RESPONSES letter, and I have no other comments. Besides, Some statements need to be improved :

As was shown in the revised Fig. 5D, the expression of PTEN α/β in the nucleus was upregulated in the A/A lane. Thus, the statement "In contrast to PTEN, PTEN α and PTEN β levels were not impacted by the S380A and S380D mutations." is inaccurate.

Reviewer #2 (Remarks to the Author):

The authors have made a strategic effort to rephrase large aspects of the manuscript in response to my initial comments. Very little experimental support has been provided to address the main concerns. My main conceptual concerns remain after this round of review, which I summarize below:

- The phosphorylation of PTEN is modelled with S to D and S to A substitutions, without accounting for the impact of the aminoacidic change on protein function beyond the phosphomimetic or phosphodead assumption. The fact that S380A and S380D have similar impact on PTEN stability already calls for caution when interpreting whether these mutants accurately model different phosphorylation states of PTEN. I believe that the results obtained in this study could be explained by molecular alterations beyond the action of aminoacidic changes in phosphorylation, and this represents an important risk to assess. Not monitoring PTEN S380 phosph in clinical samples as requested does not reduce this uncertainty.

- When asked to study PTEN S380 phosphorylation in prostate cancer, or to correlate S380 phosph with nuclear B-Cat or active B-cat signatures in gastric cancers, the authors continued to rely on an indirect estimation of PTEN phosphorylation and B-Cat activity, which reduces the conclusive capacity of the study. I restate that in order to provide a formal association between PTEN S380 phosph and Bcat nuclear loc and activity, the authors should monitor these specific parameters in clinical specimens of prostate cancer or gastric cancer, whichever is addressed experimentally. Using surrogate markers for each of the processes strongly reduces the capacity to draw any conclusions.

- PIN lesions are considered premalignant and non-cancerous. To rely on PIN as a cancer-related phenotype, when the penetrance and aggressiveness is far lower than Pten \pm mice weakens the conclusions of the model proposed. Although there are consistent differences in PIN grading as reported by the authors, PIN itself is considered a mild phenotype and not cancerous, so the difference in PIN grades would be associated to mild features that are independent from cellular transformation.

- I understand that showing an increase in activation of AKT with lower PTEN levels in PIN lesions from S380mutant mice is suggestive of additional events lowering PTEN levels. But without formal experimental proof I do not think that the authors can assume LOH. In addition, this discussion, as other aspects covered in the initial submission, distract the reader from the main discoveries that the authors pursue along the manuscript. Although the authors chose to downplay the potential LOH and the mitotic phenotype, the manuscript feels like a sum of observations with little causal demonstration, and this impacts negatively the relevance and innovative potential of the study.

- The authors chose to focus on the prostate for the S380D analysis, but the lack of data from the analysis of other tissues is an important missing aspect of the study. Especially given the fact that the authors translate clinical observations made in other tumors to modelling strategies in prostate cancer. After all, the message of the manuscript involves a primary finding in gastric cancer. As is, the study provides some clinical evidence in tumors where it lacks mechanistic analysis, and viceversa. The authors should revisit the manuscript structure to focus on a cancerous setting with

a strong clinical rationale and pursue it ambitiously.

Reviewer #3 (Remarks to the Author):

We believe the authors have sufficiently addressed our previous concerns and the manuscript is now convincing and ready for publication.

We were delighted to hear that reviewers 1 and 3 felt that their points were satisfactorily addressed and that they recommended publication of our revised manuscript. We are grateful for the continued efforts of reviewer 2 to help us further improve our study.

Below we have indicated how we have addressed the minor point of reviewer 1 and the remaining points of reviewer 2 (reviewer comments on the revised manuscript are in black font).

Response to the remaining point of reviewer 1

All the questions I concerned have been resolved in the POINT-BY-POINT RESPONSES letter, and I have no other comments. Besides, one statements need to be improved.

As was shown in the revised Fig. 5D, the expression of PTEN α/β in the nucleus was upregulated in the A/A lane. Thus, the statement "In contrast to PTEN, PTEN α and PTEN β levels were not impacted by the S380A and S380D mutations." is inaccurate.

Response: Thank you very much for re-evaluating the manuscript and for pointing out this error. We have corrected this text in the revised manuscript to incorporate this suggestion. **Please see page 7 (lines 11-13).**

Responses to the remaining points of reviewer 2

The authors have made a strategic effort to rephrase large aspects of the manuscript in response to my initial comments. Very little experimental support has been provided to address the main concerns.

Response: We would like to reaffirm our appreciation for the reviewer's feedback on the original version of the manuscript and believe that by addressing the points raised, we made several important and substantial improvements.

We regret that in two instances it was not possible to include the suggested experimental additions. Specifically, (1) the addition of a more comprehensive analysis of spontaneous tumors in *Pten*^{S380D/D} mice (beyond prostatic intraepithelial neoplasia (PIN)) would have taken approximately two years to complete considering the time needed to establish, age and analyze the necessary *Pten*^{S380D/D} and *Pten*^{+/+} cohorts of mice, and (2) experiments into the precise (epi)genetic mechanisms underlying the decline of PTEN expression from the *Pten*⁺ allele were unfortunately technically not feasible due to the small size and heterogeneous cell composition of the *Pten*^{+/-} PIN lesion.

Although we agree with the reviewer that the abovementioned experimental suggestions represent interesting extensions of our work, we also believe that the conclusions stated in the manuscript are justified based on the experimental data we now provide.

Our team has made a concerted effort to systematically address the other comments and suggestions raised by the reviewer, either by providing new experimental data or by making appropriate changes to the text. For instance, new data generated in response to comments and suggestions raised by the reviewer include:

- (1) An in-depth analysis of the mitotic phenotypes associated with the PTEN S380D substitution.
- (2) An analysis for PIN lesions in 9-month-old *Pten*^{S380D/D} and *Pten*^{+/+} males was performed to complement the 6-month-old PIN data for these strains. The added data showed that between 6 to 9 months there is a 10-fold increase in high-grade PIN III lesions in *Pten*^{S380D/D} males, with *Pten*^{+/+} males still entirely lacking such lesions. Such lesions are established precursors of prostate cancer according to a large body of literature (see below).
- (3) Inclusion of data on the potential clinical relevance of our mouse study to prostate cancer. We conducted experiments that further addressed the potential clinical relevance of our mouse studies, which was a request shared by reviewers 1 and 2. We indicated that the process for obtaining clinical samples from the designated Mayo Core Facility has been disrupted by the Covid-19 pandemic and that as a result, we were unable to conduct the precise experiments suggested by the reviewers in a timely fashion. However, in employing a commonly used alternative approach that is not dependent on tumor sections, we provided transcriptomic evidence for the potential clinical relevance of our mouse study into PTEN S380 phosphorylation.
- (4) Further analyses of potential mechanisms underlying the observed reduction in PTEN S380D and PTEN S380A levels showed that:
 - Reduced expression of these mutants is not due to reduced gene transcription or RNA stability.
 - PTEN S380D protein is less stable than WT PTEN.
 - The phosphorylation status of the C-tail residues T382, T383 and S385 is unaffected by both the S380A and S380D substitutions.

My main conceptual concerns remain after this round of review, which I summarize below:

Response: Again, we greatly appreciate the reviewer's time and effort to evaluate our revised manuscript and to provide further input. Below we have detailed our responses to each of the remaining points raised.

- The phosphorylation of PTEN is modelled with S to D and S to A substitutions, without accounting for the impact of the aminoacidic change on protein function beyond the phosphomimetic or phosphodead assumption. The fact that S380A and S380D have similar impact on PTEN stability

already calls for caution when interpreting whether these mutants accurately model different phosphorylation states of PTEN. I believe that the results obtained in this study could be explained by molecular alterations beyond the action of aminoacidic changes in phosphorylation, and this represents an important risk to assess. Not monitoring PTEN S380 phosph in clinical samples as requested does not reduce this uncertainty.

Response: The reviewer remains concerned that a loss of PTEN function/activity resulting from the S380 to D mutation rather than the acquired oncogenic property (aberrant β -catenin/WNT signaling) is the critical driver of PIN lesions in *Pten*^{S380D/D} mice. In our initial response to this concern, we reasoned that if this were to be true, then lowering β -catenin in *Pten*^{S380D/D} mice should not impact PIN lesion formation. This is not what we found, as the inactivation of a single *Ctnnb1* allele in *Pten*^{S380D/D} mice attenuated PIN lesion formation to the very low rates observed in *Pten*^{+/+} mice. Furthermore, we reasoned that if the observed loss of PTEN functions caused by the S380D-associated protein instability (e.g., failure to properly inhibit AKT phosphorylation or faithfully segregate chromosomes) were to be sufficient to drive PIN lesion formation, then one would have expected *Pten*^{S380A/A} and *Pten* ^{Δ 4/ Δ 4} mice to be PIN prone as well, which we show they are not.

That said, although we find that aberrant β -catenin/WNT signaling is critically required for PIN formation in *Pten*^{S380D/D} mice, we do agree with the reviewer that it is possible that the observed loss of PTEN functions cooperates with aberrant β -catenin/WNT signaling. We apologize for initially not recognizing this possibility offered by the reviewer and thank the reviewer for raising it again.

We discussed the reviewer's astute point regarding the potential impact of the observed partial loss of Pten functions (as reflected in the increase in PI3K-AKT signaling and mitotic errors) in the discussion section of the manuscript. **Please see page 12 (lines 19-23).**

In this context, it is also important to add that although our finding that PTEN is a tumor suppressor that can exert oncogenic properties is new, the larger concept that tumor suppressors can serve as "double agents" with tumor suppressive and oncogenic functions is not. For instance, this applies to p53 as for p53 it has been shown that several tumor-associated p53 mutants lack key tumor-suppressive functions while gaining new activities to promote tumorigenesis, as outlined in several comprehensive reviews on this topic (Freed-Pastor and Prives, 2012; Soussi and Wiman, 2015; Yue et al., 2017).

Because the abovementioned p53 findings are relevant in the context of the reviewer's point, we have included them in discussion section of the manuscript. **Please see page 12 (lines 15-19).** We anticipate that our findings on PTEN will open up a new line of inquiry in the field in years to come as has previously been the case for p53.

Additional considerations regarding the reviewer's comments about the specificity of the PTEN S380A and PTEN S380D mutants:

- i) A and D substitutions are the best available, and widely accepted, approaches to studying the different phosphorylation states of serine residues as evidenced by the following references (Chung et al., 2022; Conti et al., 2023; Gomila et al., 2022; Huang et al., 2020; Petrova et al., 2020; Xu et al., 2019). The reviewer is correct that these substitutions carry the risk that they may not accurately model a protein's actual phosphorylation states, but this risk cannot be mitigated by further experimentation.

To address the reviewer's point about the potential limitations of phospho-mimetic and phospho-dead mutants, we mentioned in the text that S->A and S->D substitutions are commonly used to model phosphorylation states but that they may not accurately do so to make readers aware of these potential limitations. **Please see text on pages 10 (lines 43-44) and 11 (lines 1-2).**

- ii) At the surface, our finding that the S380 to D mutation reduces PTEN stability seems inconsistent with what has been reported in the literature, which is a reason for concern of the reviewer. However, it is important to point out that none of the previously reported studies on PTEN C-tail phosphorylation included a PTEN S380D (or E) single mutant (as further detailed below).

Indeed, others have shown that PTEN S380A is less stable than its WT counterpart (Vasquez et al. MCB 2000; Torres and Pulido JBC 2001), which is entirely consistent with what we are finding for PTEN S380A. Importantly, to our knowledge neither these nor any other studies included the corresponding PTEN S380D (or E) mutant, so it is simply not known how the S380 phospho-mimetic mutation impacts PTEN stability and functions. Only combination D mutants of S380/T382/383 (D3) or S380/T382/383/S385 (D4) have been tested for stability but no D mutants for the individual residues. Thus, the fact that we find that PTEN S380D is less stable is not inconsistent with previously reported data on this residue.

Importantly, although the abovementioned data reported for combination D mutants suggest a more stable protein, it is important to consider that our experimentation has been conducted in vivo – under physiological conditions with endogenous mutants – whereas the abovementioned work on C-tail phosphorylation has been done in cancer cell lines with ectopically expressed PTEN variants. In the literature, there are numerous instances where the biological properties of proteins established based on protein overexpression in cancer cell lines had to be adjusted based on results obtained with genetically engineered mice or cell lines thereof.

We have incorporated the above points in the discussion section of the manuscript to point out that (1) the impact of the S380D single substitution on PTEN stability has not

been previously assessed and (2) that experimental approaches used by others were vastly distinct from the approach used by our team in the current paper. Please see text on page 10 (lines 36-43).

- iii) In making the S380D mutant mice, we obtained three independent founder lines. DNA sequencing confirmed that all three contained the anticipated nucleotide changes designed to create the S380 to D substitution. Furthermore, to minimize the risk of off-target genetic alterations introduced with CRISPR/CAS9 mediated mutagenesis that could impact PTEN function beyond the S380D substitution, we backcrossed the initial (founder) mice two times to wildtype C57BL/6 mice before intercrossing them to generate the homozygous animals. This, along with the extensive western blot analysis of PTEN S380D from *Pten*^{S380D/D} mice, suggests that it is highly unlikely that the strain contains alterations beyond the PTEN S380D substitution. Our demonstration that the level and stability of the mutant *Pten* transcript levels were normal further indicates that the amino-acidic changes that were introduced by CRISPR/CAS9 did not unintentionally perturb other parts of the *Pten* locus that could account for reduced PTEN protein levels.

We have added these details about how the *Pten*^{S380D/D} mice were generated to the methods section entitled “mouse strains”. Please see text on page 13 (lines 32-39).

- iv) Our subcellular localization data of PTEN S380A and PTEN S380D demonstrates that the two proteins possess distinct properties that are consistent with the literature and underscore that they are specific. Specifically, it has been well documented that the C-tail phosphorylation status is important for PTEN subcellular distribution. S380 is part of a nuclear exclusion motif that also includes T382, T383 and S385. Specifically, ectopic expression of PTEN-S380A, PTEN-T382A, PTEN-T383A or PTEN-S385A in U87MG cells, a glioblastoma cell line containing mutant PTEN, was shown to expose the N-terminal PTEN NLS motif, thereby driving PTEN nuclear import (each of the A mutants, including S380A, promoted PTEN nuclear import). Please see Gil et al. 2006. Our data with endogenously expressed PTEN S380A is consistent with these findings, as a much larger proportion of this mutant protein accumulates in the nucleus than with wildtype PTEN or PTEN S380D. What is striking is that even though nuclear PTEN S380D levels are much lower than those of PTEN S380A, we find nuclear PTEN S380D to abundantly interact with β -catenin whereas nuclear PTEN S380A does not, implying that phosphorylation of PTEN at S380 drives nuclear PTEN- β -catenin complex formation.

The above findings illustrate that PTEN S380A and PTEN S380D have unique properties regarding subcellular distribution and β -catenin binding, further arguing against the notion that the S380A and S380D mutants lack specificity.

We have incorporated this argument in the discussion of the paper. Please see text on pages 10 (lines 27-33).

Taken together, with the inclusion of a broad discussion of the abovementioned considerations regarding the specificity of our phospho-mimetic and phospho-dead mutants, the points raised have been comprehensively addressed.

- When asked to study PTEN S380 phosphorylation in prostate cancer, or to correlate S380 phospho with nuclear B-Cat or active B-cat signatures in gastric cancers, the authors continued to rely on an indirect estimation of PTEN phosphorylation and Bcat activity, which reduces the conclusive capacity of the study. I restate that in order to provide a formal association between PTEN S380 phospho and Bcat nuclear loc and activity, the authors should monitor these specific parameters in clinical specimens of prostate cancer or gastric cancer, whichever is addressed experimentally. Using surrogate markers for each of the processes strongly reduces the capacity to draw any conclusions.

Response: Using genetically engineered mice we uncovered that PTEN S380D targets β -catenin to exert oncogenic properties in the prostate. We then explored the potential clinical relevance of these findings focusing on gastric and prostate cancer using bioinformatics analyses on TCGA transcriptomics data. We found for both these cancer types that low expression of PDZK1 correlates with (1) hyperactivity of the β -catenin-dependent transcription factor LEF1 and (2) expression of PTEN, suggesting that our findings in mice indeed have potential clinical relevance.

Importantly, in the case of gastric cancer, others have previously demonstrated that PDZK1 binds to PTEN to inhibit phosphorylation of the PTEN S380/T382/T383 cluster, and that the PTEN C-tail and AKT are both phosphorylated at elevated levels in tumor specimens with low PDZK1 (Yang et al., 2015; Yang et al., 2013; Zhao et al., 2019). Our transcriptomic analysis of gastric cancers with low *PDZK1* transcript levels now connects these previously reported data to increased β -catenin/LEF1 target gene expression, suggesting that the oncogenic PTEN mechanism identified in PTEN S380D mice may indeed be relevant to a subset of gastric cancers.

In building further on the abovementioned PDZK1 and P-PTEN work by others, we observed that low *PDZK1* transcript levels also correlate with elevated β -catenin/LEF1- target gene expression in human prostate cancer samples, thereby providing even more support for the potential clinical relevance of the oncogenic PTEN mechanism that we identified in PTEN S380D mice. We note that if low PDZK1 expression were a poor or unreliable marker for PTEN C-tail phosphorylation, it would have been difficult to find the correlation with increased β -catenin/WNT signaling in two independent human tumor types.

While our approach to probe potential clinical relevance was not exactly as the reviewer had suggested (due to Covid-19-related difficulties in obtaining the requested clinical samples in

a timely fashion), the transcriptomic alternative approach that we used is a commonly accepted approach to probe for potential clinical relevance (Andrysik et al., 2021; Bouhaddou et al., 2020; Chen et al., 2013; Kuchay et al., 2017; Migliozzi et al., 2023; Mun et al., 2019; Shen et al., 2018). Our plan for future work on the topic of clinical translation remains to complement our initial transcriptomic evidence by performing IHC on the human prostate cancer samples that we wrote an IRB protocol for and have requested months ago. These future studies will include immunostainings for PTEN, P-PTEN S380, and β -catenin.

Unfortunately, we are still waiting for the requested samples. We have been assured that the samples will become available but the timeline for obtaining the samples remains undetermined. Because a commonly applied transcriptomic translational approach providing evidence of potential clinical relevance has already been included in the revised version of our manuscript, we sincerely hope that there will be agreement that it is reasonable to publish the manuscript without additional translational extensions. If so, we would report future IF/IHC-based experiments on the samples that we have requested in a separate, and more clinical paper.

We mentioned these future extensions in the discussion of the manuscript. Please see text on page 12 (lines 39-41).

- PIN lesions are considered premalignant and non-cancerous. To rely on PIN as a cancer-related phenotype, when the penetrance and aggressiveness is far lower than *Pten*^{+/-} mice weakens the conclusions of the model proposed. Although there are consistent differences in PIN grading as reported by the authors, PIN itself is considered a mild phenotype and not cancerous, so the difference in PIN grades would be associated to mild features that are independent from cellular transformation.

Response: Although PIN lesions are indeed not cancerous as the reviewer pointed out, these lesions are not irrelevant to the development of prostate cancer. It has been well established that PIN constitutes the neoplastic growth of epithelial cells within prostatic ducts. Low-grade PIN (I and II) are precursors of high-grade (HG) PIN (III and IV) that meet the criteria for premalignancy status and are widely accepted as precursors to prostate cancer. Please see various references regarding mouse and human PIN (Brawer, 2005; Caserta et al., 2015; Jiang et al., 2010; Majumder et al., 2004; Torres-Arzayus et al., 2004; Tosoian et al., 2018; Trotman et al., 2003; Wang et al., 2010).

Furthermore, we believe that with the addition of the 9-month-old PIN data to the 6-month-old PIN data, showing that there is a further progression to HG PIN, we had convincingly demonstrated that the lesions we are studying are relevant from a cancer perspective.

As we demonstrate in our manuscript, PIN lesion formation in *Pten*^{+/-} mice involves a reduction of PTEN expression from the remaining WT allele resulting in a robust loss of catalytic activity reminiscent of what is observed with the complete loss of PTEN (conditional *Pten*^{-/-} mice). Thus, the mechanism of transformation is different from that of *Pten*^{S380D/D} mice, which

logically underlies the difference in aggressiveness. It is important to elucidate and communicate the different mechanisms at work, further arguing that the experiments on *Pten*^{+/-} and *Pten*^{H/H} mice should be published together with our work on the C-tail mutants.

- I understand that showing an increase activation of AKT with lower PTEN levels in PIN lesions from S380mutant mice is suggestive of additional events lowering PTEN levels. But without formal experimental proof I do not think that the authors can assume LOH. In addition, this discussion, as other aspects covered in the initial submission, distract the reader from the main discoveries that the authors pursue along the manuscript. Although the authors chose to downplay the potential LOH and the mitotic phenotype, the manuscript feels like a sum of observations with little causal demonstration, and this impact negatively the relevance innovative potential of the study.

Response: In the first round of review the reviewer felt that “the analysis of the hypomorphic (mice), the different genetic backgrounds, and the LOH phenotype do not fit in the flow of the study and make the reading confusing” and asked that the data be removed and published separately. We were somewhat puzzled by this request because these studies provide important insight into the lack of tumor predisposition of *Pten*^{S380A/A} and *Pten*^{Δ4/Δ4} mice, as we articulated in the manuscript and in our response letter.

We were happy to hear that there is agreement that neoplastic growth of prostate epithelial cells in *Pten*^{+/-} mice closely correlates with a decline in PTEN expression and an increase in AKT activation (we observe this correlation at the earliest stages of transformation when lesions consist of only a few cells). We believe that knowing how PTEN levels decline to trigger neoplastic growth does not enhance the validity and novelty of these key observations, or their importance in the context of our *Pten*^{S380A/A} and *Pten*^{Δ4/Δ4} data. That said, as we indicated, we looked hard into methods to assess genetic or epigenetic LOH suitable for small lesions such as the ones we are dealing with but concluded that there aren't any. In screening the PTEN mouse literature, others have only demonstrated LOH (or the lack thereof) for a few tumor types that are relatively large and non-heterogenous, including lymphomas and mammary tumors.

Because of the above limitations, we made sure that the observed marked decline in PTEN protein in these lesions was properly portrayed as reduced level of PTEN expression from the + allele resulting in failure to inhibit the PI3K-AKT. We double-checked that the term LOH was not mentioned in the manuscript, and it was not.

- The authors chose to focus on the prostate for the S380D analysis, but the lack of data from the analysis of other tissues is an important missing aspect of the study. Especially given the fact that the authors translate clinical observations made in other tumors to modelling strategies in prostate cancer. Afterall, the message of the manuscript involves a primary finding in gastric cancer. As is, the study provides some clinical evidence in tumors where it lacks mechanistic analysis, and viceversa. The authors should revisit the manuscript structure to focus on a cancerous setting with a strong clinical rationale and pursue it ambitiously.

Response: The request for a cancer screening of *Pten*^{S380D/D} mice beyond prostate cancer was shared with reviewer 1, who asked if we could include an assessment of gastric cancer predisposition. We indicated in our response to both reviewers that we did not have the necessary mouse cohorts for these studies in place and that it would take ~2 years to complete the analysis. We still managed to be responsive to the reviewers' request of aligning mouse tumor data with clinical tumor type by correlating low PDZK1 levels in human prostate samples with PTEN expression and β -catenin/LEF1 hyperactivity.

When we started our studies seven years ago, our central goal was to decipher the role of PTEN C-tail phosphorylation at S380 in a physiological setting (at the level of the whole organism) which is the centerpiece of the manuscript. We believe our complementary assessment of the potential clinical relevance of the new mechanistic insights obtained represents a valuable added translational component.

As indicated above, we prefer to not change the manuscript structure for the reasons mentioned earlier but would welcome editorial input on this issue.

In summary, we sincerely hope that with the inclusion of a comprehensive and balanced discussion of the abovementioned topics, the reviewer agrees that the remaining comments have been sufficiently addressed.

Response of reviewer #3

We believe the authors have sufficiently addressed our previous concerns and the manuscript is now convincing and ready for publication.

We thank the reviewer for re-evaluating our manuscript and providing support for its publication.

REFERENCES

Andrysiak, Z., Bender, H., Galbraith, M.D., and Espinosa, J.M. (2021). Multi-omics analysis reveals contextual tumor suppressive and oncogenic gene modules within the acute hypoxic response. *Nat Commun* 12, 1375. 10.1038/s41467-021-21687-2.

Bouhaddou, M., Memon, D., Meyer, B., White, K.M., Rezelj, V.V., Correa Marrero, M., Polacco, B.J., Melnyk, J.E., Ulferts, S., Kaake, R.M., et al. (2020). The Global Phosphorylation Landscape of SARS-CoV-2 Infection. *Cell* 182, 685-712 e619. 10.1016/j.cell.2020.06.034.

Brawer, M.K. (2005). Prostatic intraepithelial neoplasia: an overview. *Rev Urol* 7 Suppl 3, S11-18.

Caserta, E., Egriboz, O., Wang, H., Martin, C., Koivisto, C., Pecot, T., Kladney, R.D., Shen, C., Shim, K.S., Pham, T., et al. (2015). Noncatalytic PTEN missense mutation predisposes to organ-selective cancer development in vivo. *Genes Dev* *29*, 1707-1720. 10.1101/gad.262568.115.

Chen, Y., Chi, P., Rockowitz, S., Iaquina, P.J., Shamu, T., Shukla, S., Gao, D., Sirota, I., Carver, B.S., Wongvipat, J., et al. (2013). ETS factors reprogram the androgen receptor cistrome and prime prostate tumorigenesis in response to PTEN loss. *Nat Med* *19*, 1023-1029. 10.1038/nm.3216.

Chung, S., Kang, M.S., Alimbetov, D.S., Mun, G.I., Yunn, N.O., Kim, Y., Kim, B.G., Wie, M., Lee, E.A., Ra, J.S., et al. (2022). Regulation of BRCA1 stability through the tandem UBX domains of isoleucyl-tRNA synthetase 1. *Nat Commun* *13*, 6732. 10.1038/s41467-022-34612-y.

Conti, M.M., Li, R., Narvaez Ramos, M.A., Zhu, L.J., Fazzio, T.G., and Benanti, J.A. (2023). Phosphosite Scanning reveals a complex phosphorylation code underlying CDK-dependent activation of Hcm1. *Nat Commun* *14*, 310. 10.1038/s41467-023-36035-9.

Freed-Pastor, W.A., and Prives, C. (2012). Mutant p53: one name, many proteins. *Genes Dev* *26*, 1268-1286. 10.1101/gad.190678.112.

Gomila, A.M.J., Perez-Mejias, G., Nin-Hill, A., Guerra-Castellano, A., Casas-Ferrer, L., Ortiz-Tescari, S., Diaz-Quintana, A., Samitier, J., Rovira, C., De la Rosa, M.A., et al. (2022). Phosphorylation disrupts long-distance electron transport in cytochrome c. *Nat Commun* *13*, 7100. 10.1038/s41467-022-34809-1.

Huang, H.H., Ferguson, I.D., Thornton, A.M., Bastola, P., Lam, C., Lin, Y.T., Choudhry, P., Mariano, M.C., Marcoulis, M.D., Teo, C.F., et al. (2020). Proteasome inhibitor-induced modulation reveals the spliceosome as a specific therapeutic vulnerability in multiple myeloma. *Nat Commun* *11*, 1931. 10.1038/s41467-020-15521-4.

Jiang, M., Fernandez, S., Jerome, W.G., He, Y., Yu, X., Cai, H., Boone, B., Yi, Y., Magnuson, M.A., Roy-Burman, P., et al. (2010). Disruption of PPARgamma signaling results in mouse prostatic intraepithelial neoplasia involving active autophagy. *Cell Death Differ* *17*, 469-481. 10.1038/cdd.2009.148.

Kuchay, S., Giorgi, C., Simoneschi, D., Pagan, J., Missiroli, S., Saraf, A., Florens, L., Washburn, M.P., Collazo-Lorduy, A., Castillo-Martin, M., et al. (2017). PTEN counteracts FBXL2 to promote IP3R3- and Ca(2+)-mediated apoptosis limiting tumour growth. *Nature* *546*, 554-558. 10.1038/nature22965.

Majumder, P.K., Febbo, P.G., Bikoff, R., Berger, R., Xue, Q., McMahon, L.M., Manola, J., Brugarolas, J., McDonnell, T.J., Golub, T.R., et al. (2004). mTOR inhibition reverses Akt-dependent prostate intraepithelial neoplasia through regulation of apoptotic and HIF-1-dependent pathways. *Nat Med* *10*, 594-601. 10.1038/nm1052.

Migliozzi, S., Oh, Y.T., Hasanain, M., Garofano, L., D'Angelo, F., Najac, R.D., Picca, A., Bielle, F., Di Stefano, A.L., Lerond, J., et al. (2023). Integrative multi-omics networks identify PKCdelta and DNA-PK as master kinases of glioblastoma subtypes and guide targeted cancer therapy. *Nat Cancer*. 10.1038/s43018-022-00510-x.

Mun, D.G., Bhin, J., Kim, S., Kim, H., Jung, J.H., Jung, Y., Jang, Y.E., Park, J.M., Kim, H., Jung, Y., et al. (2019). Proteogenomic Characterization of Human Early-Onset Gastric Cancer. *Cancer Cell* 35, 111-124 e110. 10.1016/j.ccell.2018.12.003.

Petrova, V., Pearson, C.S., Ching, J., Tribble, J.R., Solano, A.G., Yang, Y., Love, F.M., Watt, R.J., Osborne, A., Reid, E., et al. (2020). Protrudin functions from the endoplasmic reticulum to support axon regeneration in the adult CNS. *Nat Commun* 11, 5614. 10.1038/s41467-020-19436-y.

Shen, S.M., Ji, Y., Zhang, C., Dong, S.S., Yang, S., Xiong, Z., Ge, M.K., Yu, Y., Xia, L., Guo, M., et al. (2018). Nuclear PTEN safeguards pre-mRNA splicing to link Golgi apparatus for its tumor suppressive role. *Nat Commun* 9, 2392. 10.1038/s41467-018-04760-1.

Soussi, T., and Wiman, K.G. (2015). TP53: an oncogene in disguise. *Cell Death Differ* 22, 1239-1249. 10.1038/cdd.2015.53.

Torres-Arzuayus, M.I., Font de Mora, J., Yuan, J., Vazquez, F., Bronson, R., Rue, M., Sellers, W.R., and Brown, M. (2004). High tumor incidence and activation of the PI3K/AKT pathway in transgenic mice define AIB1 as an oncogene. *Cancer Cell* 6, 263-274. 10.1016/j.ccr.2004.06.027.

Tosoian, J.J., Alam, R., Ball, M.W., Carter, H.B., and Epstein, J.I. (2018). Managing high-grade prostatic intraepithelial neoplasia (HGPIN) and atypical glands on prostate biopsy. *Nat Rev Urol* 15, 55-66. 10.1038/nrrol.2017.134.

Trotman, L.C., Niki, M., Dotan, Z.A., Koutcher, J.A., Di Cristofano, A., Xiao, A., Khoo, A.S., Roy-Burman, P., Greenberg, N.M., Van Dyke, T., et al. (2003). Pten dose dictates cancer progression in the prostate. *PLoS Biol* 1, E59. 10.1371/journal.pbio.0000059.

Wang, H., Karikomi, M., Naidu, S., Rajmohan, R., Caserta, E., Chen, H.Z., Rawahneh, M., Moffitt, J., Stephens, J.A., Fernandez, S.A., et al. (2010). Allele-specific tumor spectrum in pten knockin mice. *Proc Natl Acad Sci U S A* 107, 5142-5147. 10.1073/pnas.0912524107.

Xu, W., Beebe, K., Chavez, J.D., Boysen, M., Lu, Y., Zuehlke, A.D., Keramisanou, D., Trepel, J.B., Prodromou, C., Mayer, M.P., et al. (2019). Hsp90 middle domain phosphorylation initiates a complex conformational program to recruit the ATPase-stimulating cochaperone Aha1. *Nat Commun* 10, 2574. 10.1038/s41467-019-10463-y.

Yang, Z., Xie, C., Xu, W., Liu, G., Cao, X., Li, W., Chen, J., Zhu, Y., Luo, S., Luo, Z., and Lu, N. (2015). Phosphorylation and inactivation of PTEN at residues Ser380/Thr382/383 induced by *Helicobacter pylori* promotes gastric epithelial cell survival through PI3K/Akt pathway. *Oncotarget* 6, 31916-31926. 10.18632/oncotarget.5577.

Yang, Z., Yuan, X.G., Chen, J., Luo, S.W., Luo, Z.J., and Lu, N.H. (2013). Reduced expression of PTEN and increased PTEN phosphorylation at residue Ser380 in gastric cancer tissues: a novel mechanism of PTEN inactivation. *Clin Res Hepatol Gastroenterol* 37, 72-79. 10.1016/j.clinre.2012.03.002.

Yue, X., Zhao, Y., Xu, Y., Zheng, M., Feng, Z., and Hu, W. (2017). Mutant p53 in Cancer: Accumulation, Gain-of-Function, and Therapy. *J Mol Biol* 429, 1595-1606. 10.1016/j.jmb.2017.03.030.

Zhao, C., Tao, T., Yang, L., Qin, Q., Wang, Y., Liu, H., Song, R., Yang, X., Wang, Q., Gu, S., et al. (2019). Loss of PDZK1 expression activates PI3K/AKT signaling via PTEN phosphorylation in gastric cancer. *Cancer Lett* 453, 107-121. 10.1016/j.canlet.2019.03.043.